# SCALE MIXTURES OF NEURAL NETWORK GAUSSIAN PROCESSES

**Hyungi Lee**[1]**, Eunggu Yun**[1]**, Hongseok Yang**[1,2,3]**, Juho Lee**[1,4]
[1]Kim Jaechul Graduate School of AI, KAIST, South Korea
[2]School of Computing, KAIST, South Korea
[3]Discrete Mathematics Group, Institute for Basic Science (IBS), Daejeon, South Korea
[4]AITRICS, Seoul, South Korea
{lhk2708, eunggu.yun, hongseok.yang, juholee}@kaist.ac.kr

## ABSTRACT

Recent works have revealed that infinitely-wide feed-forward or recurrent neural networks of any architecture correspond to Gaussian processes referred to as Neural Network Gaussian Processes (NNGPs). While these works have extended the class of neural networks converging to Gaussian processes significantly, however, there has been little focus on broadening the class of stochastic processes that such neural networks converge to. In this work, inspired by the scale mixture of Gaussian random variables, we propose the scale mixture of NNGPs for which we introduce a prior distribution on the scale of the last-layer parameters. We show that simply introducing a scale prior on the last-layer parameters can turn infinitely-wide neural networks of any architecture into a richer class of stochastic processes. With certain scale priors, we obtain heavy-tailed stochastic processes, and in the case of inverse gamma priors, we recover Student's $t$ processes. We further analyze the distributions of the neural networks initialized with our prior setting and trained with gradient descents and obtain similar results as for NNGPs. We present a practical posterior-inference algorithm for the scale mixture of NNGPs and empirically demonstrate its usefulness on regression and classification tasks. In particular, we show that in both tasks, the heavy-tailed stochastic processes obtained from our framework are robust to out-of-distribution data.

## 1 INTRODUCTION

There has been growing interest in the literature on the equivalence between wide deep neural networks and Gaussian Processes (GPs). Neal (1996) showed that a shallow but infinitely-wide Bayesian Neural Network (BNN) with random weights and biases corresponds to a GP. This result was extended to fully-connected deep neural networks of any depth (Lee et al., 2018; Matthews et al., 2018), which are shown to converge to GPs as the width grows. Similar results were later obtained for deep Convolutional Neural Networks (CNNs) (Novak et al., 2018; Garriga-Alonso et al., 2019) and attention networks (Hron et al., 2020). In fact, Yang (2019) showed that wide feed-forward or recurrent neural networks of *any* architecture converge to GPs and presented a generic method for computing kernels for such GPs. Under this correspondence, the posterior inference of an infinitely-wide BNN boils down to the posterior inference of the corresponding GP called NNGP for which a closed-form posterior can be computed exactly.

Our goal is to advance this line of research by going beyond GPs. We present a simple yet flexible recipe for constructing infinitely-wide BNNs that correspond to a wide range of stochastic processes. Our construction includes *heavy-tailed* stochastic processes such as Student's $t$ processes which have been demonstrated to be more robust than GPs under certain scenarios (Shah et al., 2014).

Our construction is inspired by a popular class of distributions called *scale mixtures of Gaussians* (Andrews & Mallows, 1974); such a distribution is obtained by putting a prior on the scale or variance parameter of a Gaussian distribution. We extend this scale mixing to NNGPs, where we introduce a prior distribution on the scale of the parameters for the *last layer* (which is often referred to as *readout layer*) in a wide neural network. We show that simply introducing a scale prior

on the last layer can turn infinitely-wide BNNs of any architecture into a richer class of stochastic processes, which we name as *scale mixtures of* NNGPs. These scale mixtures include a broad class of stochastic processes, such as heavy-tailed processes which are shown to be robust to outliers. In particular, when the prior on the scale is inverse gamma, the scale mixture of NNGPs becomes Stduent's $t$ process (Shah et al., 2014).

We demonstrate that, despite increasing flexibility, mixing NNGPs with a prior on the scale parameter does not increase the difficulty of posterior inference much or at all in some cases. When we mix the scale parameter with inverse gamma (so that the mixture becomes a $t$ process), we can compute the kernel efficiently and infer the exact posterior, as in the case of NNGPs. For generic scale priors and regression tasks with them, we present an efficient approximate posterior-inference algorithm based on importance sampling, which saves computation time by reusing the shared covariance kernels. For classification tasks with categorical likelihood (for which an exact posterior is not available even for the original NNGP), we present an efficient stochastic variational inference algorithm.

We further analyze the distributions of the neural networks initialized with our prior setting and trained with gradient descents and obtain results similar to the ones for NNGPs. For NNGP, it has been shown (Matthews et al., 2017; Lee et al., 2019) that when a wide neural network is initialized with the NNGP specification and then trained only for the last layer (with all the other layers fixed), its fully trained version becomes a random function drawn from the NNGP posterior. Similarly, we analyze the distribution of a wide neural network of any architecture initialized with our prior specification and trained only for the last layer, and show that it becomes a sample from a scale mixture of NNGPs with some scaling distribution. Interestingly, the scaling distribution is a prior, not a posterior. For the fully-connected wide neural networks, we extend the analysis to the case where all the layers are trained with gradient descent, and show that the limiting distribution is again a scale mixture of GPs with a prior scaling distribution and each GPs using a kernel called Neural Tangent Kernel (NTK). In the case of an inverse gamma prior, the limiting distribution becomes Student's $t$ process, which can be computed analytically.

We empirically show the usefulness of our construction on various real-world regression and classification tasks. We demonstrate that, despite the increased flexibility, the scale mixture of NNGPs is readily applicable to most of the problems where NNGPs are used, without increasing the difficulty of inference. Moreover, the heavy-tailed processes derived from our construction are shown to be more robust than NNGPs for out-of-distribution or corrupted data while maintaining similar performance for the normal data. Our empirical analysis suggests that our construction is not merely a theoretical extension of the existing framework, but also provides a practical alternative to NNGPs.

## 1.1 RELATED WORKS

**NNGP and NTK**  Our construction heavily depends on the tensor-program framework (Yang, 2019) which showed that a wide BNN of any feed-forward or recurrent architecture converges in distribution to an NNGP, as its width increases. Especially, we make use of the so-called master theorem and its consequence (reviewed in Section 1.1) to derive our results. Building on the seminal work on NTK (Jacot et al., 2018), Lee et al. (2019) analyzed the dynamics of fully-connected neural networks trained with gradient descent and showed that fully-trained infinitely-wide networks are GPs. In particular, they confirmed the result in Matthews et al. (2017) that when a fully-connected neural network is initialized from a specific prior (often referred to as the NTK parameterization) and trained only for the last layer under gradient descent and squared loss, its fully-trained version becomes a posterior sample of the corresponding NNGP. Lee et al. (2019) further analyzed the case where all the layers are trained with gradient descent and showed that the network also converges to a GP with specific parameters computed with the so called NTK kernel. We extend these results to our scale mixture of NNGPs in Section 3.

**Heavy-tailed stochastic processes from infinitely-wide BNNs**  The attempts to extend the results on NNGPs to heavy-tailed processes have been made in the past, although not common. The representative case is the work by Favaro et al. (2020), which showed that under an alternative prior specification, a wide fully-connected neural network converges to stable processes, as the widths increase. This result was later extended to deep CNNs in (Bracale et al., 2021). What distinguishes our approach from these works is the simplicity of our construction; it simply puts a prior distribution on the scale of the last-layer parameters of a network and makes the scale a random variable, while

the constructions in those works replace priors for entire network parameters from Gaussian to other distributions. This simplicity has multiple benefits. First, most of the nice properties of NNGPs, such as easy-to-compute posteriors and the correspondence to gradient descent training, continue to hold in our approach as a version for mixture distributions, while it is at least difficult to find similar adjustments of these results in those works. Second, our approach is applicable to arbitrary architectures, while those works considered only fully-connected networks and CNNs.

## 2 PRELIMINARIES

We start with a quick review on a few key concepts used in our results. For $M \in \mathbb{N}$, let $[M]$ be the set $\{1, \ldots, M\}$. Also, write $\mathbb{R}_+$ for the set $\{x \in \mathbb{R} \mid x > 0\}$.

### 2.1 TENSOR PROGRAMS AND NEURAL NETWORK GAUSSIAN PROCESSES

Tensor programs (Yang, 2019) are a particular kind of straight-line programs that express computations of neural networks on fixed inputs. In a sense, they are similar to computation graphs used by autodiff packages, such as TensorFlow and PyTorch, but differ from computation graphs in two important aspects. First, a single tensor program can express the computations of infinitely-many neural networks, which share the same architecture but have different widths. This capability comes from the parameter $n$ of the tensor program, which determines the dimensions of vectors and matrices used in the program. The parameter corresponds to the width of the hidden layers of a neural network, so that changing $n$ makes the same tensor program model multiple neural networks of different widths. Second, tensor programs describe stochastic computations, where stochasticity comes from the random initialization of network weights and biases. These two points mean that we can use a single tensor program to model the sequence of BNNs of the same architecture but with increasing widths, and understand the limit of the sequence by analyzing the program. The syntax of and other details about tensor programs are given in Appendix A.

We use tensor programs because of the so called master theorem, which provides a method for computing the infinite-width limits of BNN sequences expressed by tensor programs. For the precise formulation of the theorem and the method, again see Appendix A.

To explain the master theorem more precisely, assume that we are given a family of BNNs $\{f_n\}_{n \in N}$ that are indexed by their widths $n$ and have the following form:

$$f_n(-; \mathbf{v}_n, \mathbf{\Psi}_n) : \mathbb{R}^I \to \mathbb{R}, \qquad f_n(\mathbf{x}; \mathbf{v}_n, \mathbf{\Psi}_n) = \frac{1}{\sqrt{n}} \sum_{\alpha \in [n]} \mathbf{v}_{n,\alpha} \cdot \phi(g_n(\mathbf{x}; \mathbf{\Psi}_n))_\alpha,$$

where $\mathbf{v}_n \in \mathbb{R}^n$ and $\mathbf{\Psi}_n \in \mathbb{R}^{n \times P}$ (for some fixed $P$) are the parameters of the $n$-th network, the former being the ones of the readout layer and the latter all the other parameters, $I$ is the dimension of the inputs, $\phi$ is a non-linear activation function of the network, and $g_n(\mathbf{x}; \mathbf{\Psi}_n)$ is the output of a penultimate linear layer, which feeds directly to the readout layer after being transformed by the activation function. In order for the computations of these BNNs on some inputs to be represented by a tensor program, the components of the networks have to satisfy at least the following two conditions. First, the entries of $\mathbf{v}_n$ and $\mathbf{\Psi}_n$ are initialized with samples drawn independently from (possibly different) zero-mean Gaussian distributions. For the entries of $\mathbf{v}_n$, the distributions are the same and have the variance $\sigma_v^2$ for some $\sigma_v > 0$. For the entries of $\mathbf{\Psi}_n$, the distributions may be different, but if we restrict them to those in each column $j$ of $\mathbf{\Psi}_n$, they become the same and have the variance $\sigma_j/n$ for some $\sigma_j > 0$. Second, $\phi(z)$ is *controlled*, i.e., it is bounded by a function $\exp(C|z|^{2-\epsilon} + c)$ for some $C, \epsilon, c > 0$. This property ensures that $\phi(z)$ for any Gaussian variable $z$ has a finite expectation.

Let $\mathbf{x}_1, \ldots, \mathbf{x}_M \in \mathbb{R}^I$ be $M$ inputs. When the computations of the BNNs on these inputs are represented by a single tensor program, the master theorem holds. It says that there is a general method for computing $\mu \in \mathbb{R}^M$ and $\Sigma \in \mathbb{R}^{M \times M}$ inductively on the syntax of the program such that $\mu$ and $\Sigma$ characterize the limit of $(g_n(\mathbf{x}_1; \mathbf{\Psi}_n), \ldots, g_n(\mathbf{x}_M; \mathbf{\Psi}_n))_{n \in \mathbb{N}}$ in the following sense.

**Theorem 2.1** (Master Theorem). *Let $h : \mathbb{R}^M \to \mathbb{R}$ be a controlled function. Then, as $n$ tends to $\infty$,*

$$\frac{1}{n} \sum_{\alpha=1}^{n} h\Big(g_n(\mathbf{x}_1; \mathbf{\Psi}_n)_\alpha, \ldots, g_n(\mathbf{x}_M; \mathbf{\Psi}_n)_\alpha\Big) \xrightarrow{a.s.} \mathbb{E}_{Z \sim \mathcal{N}(\mu, \Sigma)} [h(Z)], \tag{1}$$

*where* $\xrightarrow{a.s.}$ *refers to almost sure convergence.*

The next corollary describes an important consequence of the theorem.

**Corollary 2.2.** *As $n$ tends to $\infty$, the joint distribution of $(f_n(\mathbf{x}_m; \mathbf{v}_n, \mathbf{\Psi}_n))_{m \in [M]}$ converges weakly to the multivariate Gaussian distribution with zero mean and the following covariance matrix $\mathcal{K} \in \mathbb{R}^{M \times M}$:*

$$\mathcal{K}_{(i,j)} = \sigma_v^2 \cdot \mathbb{E}_{Z \sim \mathcal{N}(\mu, \Sigma)} \left[ \phi(Z_i) \phi(Z_j) \right] \tag{2}$$

*where $Z_i$ and $Z_j$ are the $i$-th and $j$-th components of $Z$.*

The above corollary and the fact that $\mathcal{K}$ depend only on the shared architecture of the BNNs, not on the inputs $\mathbf{x}_{1:M}$, imply that the BNNs converge to a GP whose mean is the constant zero function and whose kernel is

$$\kappa(\mathbf{x}, \mathbf{x}') = \sigma_v^2 \cdot \mathbb{E}_{Z \sim \mathcal{N}(\mu, \Sigma)} \left[ \phi(Z_1) \phi(Z_2) \right] \tag{3}$$

where $(\mu, \Sigma)$ is constructed as described above for the case that $M = 2$ and $(\mathbf{x}_1, \mathbf{x}_2) = (\mathbf{x}, \mathbf{x}')$. This GP is called *Neural Network Gaussian Process* (NNGP) in the literature.

## 2.2 STUDENT'S $t$ PROCESSES

Student's $t$ processes feature prominently in our results to be presented shortly, because they are marginals of the scale mixtures of NNGPs when the mixing is done by inverse gamma. These processes are defined in terms of multivariate Student's $t$ distributions defined as follows.

**Definition 2.1.** A distribution on $\mathbb{R}^d$ is a *multivariate (Student's) $t$ distribution* with degree of freedom $\nu \in \mathbb{R}_+$, location $\mu \in \mathbb{R}^d$ and positive-definite scale matrix $\Sigma \in \mathbb{R}^{d \times d}$ if it has the following density:

$$p(\mathbf{x}) = \frac{G((\nu + d)/2)}{(\nu\pi)^{\frac{d}{2}} \cdot G(\nu/2) \cdot |\Sigma|^{\frac{1}{2}}} \left( 1 + \frac{1}{\nu}(\mathbf{x} - \mu)^\top \Sigma^{-1}(\mathbf{x} - \mu) \right)^{-\frac{\nu+d}{2}}$$

where $G(\cdot)$ is the gamma function. To express a random variable drawn from this distribution, we write $\mathbf{x} \sim \mathrm{MVT}_d(\nu, \mu, \Sigma)$.

When $\nu > 1$, the mean of the distribution $\mathrm{MVT}_d(\nu, \mu, \Sigma)$ exists, and it is $\mu$. When $\nu > 2$, the covariance of distribution also exists, and it is $\frac{\nu}{\nu-2}\Sigma$. As for the Gaussian distribution, the multivariate $t$ distribution is also closed under marginalization and conditioning. More importantly, it can be obatined as a scale mixture of Gaussians; if $\sigma^2 \sim \mathrm{InvGamma}(a, b)$ and $\mathbf{x}|\sigma^2 \sim \mathcal{N}(\mu, \sigma^2\Sigma)$, then the marginal distribution of $\mathbf{x}$ is $\mathrm{MVT}(2a, \mu, \frac{b}{a}\Sigma)$. For the definition of the inverse gamma distribution, see Definition B.1.

Student's $t$ process is a real-valued random function such that the outputs of the function on a finite number of inputs are distributed by a multivariate $t$ distribution (Shah et al., 2014). Consider a set $\mathcal{X}$ (for inputs) with a symmetric positive-definite function $\kappa : \mathcal{X} \times \mathcal{X} \to \mathbb{R}$.

**Definition 2.2.** A random function $f : \mathcal{X} \to \mathbb{R}$ is *Student's $t$ process* with degree of freedom $\nu \in \mathbb{R}_+$, location function $M : \mathcal{X} \to \mathbb{R}$, and scale function $\kappa$, if for all $d$ and inputs $x_1, \ldots, x_d \in \mathcal{X}$, the random vector $(f(x_1), \ldots, f(x_d))$ has the multivariate $t$ distribution $\mathrm{MVT}_d(\nu, \mu, \Sigma)$, where $\mu = (M(x_1), \ldots, M(x_d))$ and $\Sigma$ is the $d \times d$ matrix defined by $\Sigma_{(i,j)} = k(x_i, x_j)$.

## 3 RESULTS

We will present our results for the regression setting. Throughout this section, we consider only those BNNs whose computations over fixed inputs are representable by tensor programs. Assume that we are given a training set $\mathcal{D}_{\mathrm{tr}} = \{(\mathbf{x}_k, y_k) \in \mathbb{R}^I \times \mathbb{R} \mid 1 \leq k \leq K\}$ and a test set $\mathcal{D}_{\mathrm{te}} = \{(\mathbf{x}_{K+\ell}, y_{K+\ell}) \mid 1 \leq \ell \leq L\}$.

Our idea is to treat the variance $\sigma_v^2$ for the parameters of the readout layer in a BNN as a random variable, not a deterministic value, so that the BNN represents a mixture of random functions (where the randomness comes from that of $\sigma_v^2$ and the random initialization of the network parameters). To

state this more precisely, consider the computations of the BNNs $\{f_n(-; \mathbf{v}_n, \mathbf{\Psi}_n)\}_{n \in \mathbb{N}}$ on the inputs $\mathbf{x}_{1:K+L}$ in the training and test sets such that $n$ is the width of the hidden layers of $f_n$ and these computations are representable by a tensor program. Our idea is to change the distribution for the initial value of $\mathbf{v}_n$ from a Gaussian to a scale mixture of Gaussians, so that at initialization, the BNNs on $\mathbf{x}_{1:K+L}$ become the following random variables: for each $n \in \mathbb{N}$,

$$\sigma_v^2 \sim \mathcal{H}, \qquad \mathbf{v}_{n,\alpha} | \sigma_v^2 \sim \mathcal{N}(0, \sigma_v^2) \text{ for } \alpha \in [n], \qquad \mathbf{\Psi}_{n,(\alpha,j)} \sim \mathcal{N}(0, \sigma_j^2/n) \text{ for } \alpha \in [n], j \in [P],$$

$$f_n(\mathbf{x}_i; \mathbf{v}_n, \mathbf{\Psi}_n) = \frac{1}{\sqrt{n}} \sum_{\alpha \in [n]} \mathbf{v}_{n,\alpha} \cdot \phi(g_n(\mathbf{x}_i; \mathbf{\Psi}_n))_\alpha \text{ for } i \in [K+L],$$

where $\mathcal{H}$ is a distribution on positive real numbers, such as the inverse-gamma distribution, $P$ is the number of columns of $\mathbf{\Psi}_n$, and $\sigma_j^2$ and $g_n$ are, respectively, the variance specific to the $j$-th column of $\mathbf{\Psi}_n$ and the output of the penultimate linear layer, as explained in our review on tensor programs.

Our first result says that as the width of the BNN grows to $\infty$, the BNN converges in distribution to a mixture of NNGPs, and, in particular, to Student's $t$ process if the distribution $\mathcal{H}$ is inverse gamma.

**Theorem 3.1** (Convergence). *As $n$ tends to $\infty$, the random variable $(f_n(\mathbf{x}_i; \mathbf{v}_n, \mathbf{\Psi}_n))_{i \in [K+L]}$ converges in distribution to the following random variable $(f_\infty(\mathbf{x}_i))_{i \in [K+L]}$:*

$$\sigma_v^2 \sim \mathcal{H}, \qquad (f_\infty(\mathbf{x}_i))_{i \in [K+L]} | \sigma_v^2 \sim \mathcal{N}(0, \sigma_v^2 \cdot \overline{\mathcal{K}}),$$

*where $\overline{\mathcal{K}}_{(i,j)} = \mathbb{E}_{Z \sim \mathcal{N}(\mu, \Sigma)}[\phi(Z_i)\phi(Z_j)]$ for $i, j \in [K+L]$. Furthermore, if $\mathcal{H}$ is the inverse-gamma distribution with shape $a$ and scale $b$, the marginal distribution of $(f_\infty(\mathbf{x}_i))_{i \in [K+L]}$ is the following multivariate $t$ distribution:*

$$(f_\infty(\mathbf{x}_i))_{i \in [K+L]} \sim \mathrm{MVT}_{K+L}(2a, 0, (b/a) \cdot \overline{\mathcal{K}}),$$

*where $\mathrm{MVT}_d(\nu', \mu', \mathcal{K}')$ denotes the $d$-dimensional Student's $t$ distribution with degree of freedom $\nu'$, location $\mu'$, and scale matrix $\mathcal{K}'$.*

This theorem holds mainly because of the master theorem for tensor programs (Theorem 2.1). In fact, the way that its proof uses the master theorem is essentially the same as the one of the NNGP convergence proof (i.e., the proof of Corollary 2.2), except for one thing: before applying the master theorem, the proof of Theorem 3.1 conditions on $\sigma_v^2$ and removes its randomness. The detailed proof can be found in the appendix.

As in the case of Corollary 2.2, although the theorem is stated for the finite dimensional case with the inputs $\mathbf{x}_{1:K+L}$, it implies the convergence of the BNNs with random $\sigma_v^2$ to a scale mixture of GPs or to Student's $t$ process. Note that $\sigma_v^2 \cdot \overline{\mathcal{K}}$ in the theorem is precisely the covariance matrix $\mathcal{K}$ of the NNGP kernel on $\mathbf{x}_{1:K+L}$ in Corollary 2.2. We thus call the limiting stochastic process as *scale mixture of* NNGPs.

Our next results characterize a fully-trained BNN at the infinite-width limit under two different training schemes. They are the versions of the standard results in the literature, generalized from GPs to scale mixtures of GPs. Assume the BNNs under the inputs $\mathbf{x}_{1:K+L}$ that we have been using so far.

**Theorem 3.2** (Convergence under Readout-Layer Training). *Assume that we train only the readout layers of the BNNs using the training set $\mathcal{D}_{tr}$ under mean squared loss and infinitesimal step size. Formally, this means the network parameters are evolved under the following differential equations:*

$$(\mathbf{v}_n^{(0)}, \mathbf{\Psi}_n^{(0)}) = (\mathbf{v}_n, \mathbf{\Psi}_n), \quad \frac{d\mathbf{v}_n^{(t)}}{dt} = \sum_{i=1}^K \left( y_i - f_n(\mathbf{x}_i; \mathbf{v}_n^{(t)}, \mathbf{\Psi}_n^{(t)}) \right) \frac{2\phi(g_n(\mathbf{x}_i; \mathbf{\Psi}_n^{(t)}))}{K\sqrt{n}}, \quad \frac{d\mathbf{\Psi}_n^{(t)}}{dt} = 0.$$

*Then, the time and width limit of the random variables $(f_n(\mathbf{x}_{K+i}; \mathbf{v}_n^{(t)}, \mathbf{\Psi}_n^{(t)}))_{i \in [L]}$, denoted $(f_\infty^\infty(\mathbf{x}_{K+i}))_{i \in [L]}$, is distributed by the following mixture of Gaussians:*

$$\sigma_v^2 \sim \mathcal{H}, \qquad (f_\infty^\infty(\mathbf{x}_{K+i}))_{i \in [L]} | \sigma_v^2 \sim \mathcal{N}\left( \left( \overline{\mathcal{K}}_{te,tr} \overline{\mathcal{K}}_{tr,tr}^{-1} Y_{tr} \right), \sigma_v^2 \left( \overline{\mathcal{K}}_{te,te} - \overline{\mathcal{K}}_{te,tr} \overline{\mathcal{K}}_{tr,tr}^{-1} \overline{\mathcal{K}}_{tr,te} \right) \right),$$

*where $Y_{tr}$ consists of the $y$ values in $\mathcal{D}_{tr}$ (i.e., $Y_{tr} = (y_1, \ldots, y_K)$) and the $\overline{\mathcal{K}}$'s with different subscripts are the restrictions of $\overline{\mathcal{K}}$ in Theorem 3.1 with training or test inputs as directed by those subscripts. Furthermore, if $\mathcal{H}$ is the inverse-gamma distribution with shape $a$ and scale $b$, the marginal distribution of $(f_\infty^\infty(\mathbf{x}_{K+i}))_{i \in [L]}$ is the following multivariate $t$ distribution:*

$$(f_\infty^\infty(\mathbf{x}_{K+i}))_{i \in [L]} \sim \mathrm{MVT}_L\left( 2a, \left( \overline{\mathcal{K}}_{te,tr} \overline{\mathcal{K}}_{tr,tr}^{-1} Y_{tr} \right), \frac{b}{a} \left( \overline{\mathcal{K}}_{te,te} - \overline{\mathcal{K}}_{te,tr} \overline{\mathcal{K}}_{tr,tr}^{-1} \overline{\mathcal{K}}_{tr,te} \right) \right).$$

One important point about this theorem is that the distribution of $(f_\infty^\infty(\mathbf{x}_{K+i}))_{i\in[L]}$ is not the posterior of $(f_\infty(\mathbf{x}_{K+i}))_{i\in[L]}$ in Theorem 3.1 under the condition on $Y_{\mathrm{tr}}$. The former uses the prior on $\sigma_v^2$ for mixing the distribution of $(f_\infty(\mathbf{x}_{K+i}))_{i\in[L]} \mid (\sigma_v^2, Y_{\mathrm{tr}})$, while the latter uses the posterior on $\sigma_v^2$ for the same purpose.

Our last result assumes a change in the initialization rule for our models. This is to enable the use of the results in Lee et al. (2019) about the training of infinitely-wide neural networks. Concretely, we use the so called NTK parametrization for $\boldsymbol{\Psi}_n$. This means that the BNNs on $\mathbf{x}_{1:K+L}$ are now intialized to be the following random variables: for each $n \in \mathbb{N}$,

$$\sigma_v^2 \sim \mathcal{H}, \quad \mathbf{v}_{n,\alpha}|\sigma_v^2 \sim \mathcal{N}(0, \sigma_v^2), \quad \boldsymbol{\Psi}_{n,(\alpha,j)} \sim \mathcal{N}(0, \sigma_j^2) \quad \text{for } \alpha \in [n], j \in [P],$$

$$f_n(\mathbf{x}_i; \mathbf{v}_n, \frac{1}{\sqrt{n}}\boldsymbol{\Psi}_n) = \frac{1}{\sqrt{n}} \sum_{\alpha\in[n]} \mathbf{v}_{n,\alpha} \cdot \phi(g_n(\mathbf{x}_i; \frac{1}{\sqrt{n}}\boldsymbol{\Psi}_n))_\alpha \quad \text{for } i \in [K+L].$$

When we adopt this NTK parameterization and the BNNs are multi-layer perceptrons (more generally, they share an architecture for which the limit theory of neural tangent kernels has been developed), we can analyze their distributions after the training of all parameters. Our analysis uses the following famous result for such BNNs: as $n$ tends to infinity, for all $i, j \in [K+L]$,

$$\left\langle \nabla_{(\mathbf{v}_n, \boldsymbol{\Psi}_n)} f_n(\mathbf{x}_i; \mathbf{v}_n, \frac{1}{\sqrt{n}}\boldsymbol{\Psi}_n), \nabla_{(\mathbf{v}_n, \boldsymbol{\Psi}_n)} f_n(\mathbf{x}_j; \mathbf{v}_n, \frac{1}{\sqrt{n}}\boldsymbol{\Psi}_n) \right\rangle \xrightarrow{a.s.} \Theta_{(i,j)} \tag{4}$$

for some $\Theta_{(i,j)}$ determined by the architecture of the BNNs. Let $\Theta$ be the matrix $(\Theta_{(i,j)})_{i,j\in[K+L]}$. The next theorem describes the outcome of our analysis for the fully-trained infinite-width BNN.

**Theorem 3.3** (Convergence under General Training). *Assume that the BNNs are multi-layer perceptrons and we train all of their parameters using the training set $\mathcal{D}_{tr}$ under mean squared loss and infinitesimal step size. Formally, this means the network parameters are evolved under the following differential equations:*

$$\mathbf{v}_n^{(0)} = \mathbf{v}_n, \qquad \frac{d\mathbf{v}_n^{(t)}}{dt} = \frac{2}{K\sqrt{n}} \sum_{i=1}^{K} \left( y_i - f_n(\mathbf{x}_i; \mathbf{v}_n^{(t)}, \frac{1}{\sqrt{n}}\boldsymbol{\Psi}_n^{(t)}) \right) \phi(g_n(\mathbf{x}_i; \frac{1}{\sqrt{n}}\boldsymbol{\Psi}_n^{(t)})),$$

$$\boldsymbol{\Psi}_n^{(0)} = \boldsymbol{\Psi}_n, \qquad \frac{d\boldsymbol{\Psi}_n^{(t)}}{dt} = \frac{2}{K} \sum_{i=1}^{K} \left( y_i - f_n(\mathbf{x}_i; \mathbf{v}_n^{(t)}, \frac{1}{\sqrt{n}}\boldsymbol{\Psi}_n^{(t)}) \right) \nabla_{\boldsymbol{\Psi}} f_n(\mathbf{x}_i; \mathbf{v}_n^{(t)}, \frac{1}{\sqrt{n}}\boldsymbol{\Psi}_n^{(t)}).$$

*Let $\bar{\Theta}$ be $\Theta$ with $\sigma_v^2$ in it set to 1 (so that $\Theta = \sigma_v^2\bar{\Theta}$). Then, the time and width limit of the random variables $(f_n(\mathbf{x}_{K+i}; \mathbf{v}_n^{(t)}, \frac{1}{\sqrt{n}}\boldsymbol{\Psi}_n^{(t)}))_{i\in[L]}$ has the following distribution:*

$$\sigma_v^2 \sim \mathcal{H}, \qquad\qquad (f_\infty^\infty(\mathbf{x}_{K+i}))_{i\in[L]}|\sigma_v^2 \sim \mathcal{N}(\mu', \sigma_v^2\Theta')$$

*where*

$$\mu' = \bar{\Theta}_{te,tr}\bar{\Theta}_{tr,tr}^{-1}Y_{tr},$$

$$\Theta' = \overline{\mathcal{K}}_{te,te} + \left( \bar{\Theta}_{te,tr}\bar{\Theta}_{tr,tr}^{-1}\overline{\mathcal{K}}_{tr,tr}\bar{\Theta}_{tr,tr}^{-1}\bar{\Theta}_{tr,te} \right) - \left( \bar{\Theta}_{te,tr}\bar{\Theta}_{tr,tr}^{-1}\overline{\mathcal{K}}_{tr,te} + \overline{\mathcal{K}}_{te,tr}\bar{\Theta}_{tr,tr}^{-1}\bar{\Theta}_{tr,te} \right).$$

*Here the $\overline{\mathcal{K}}$'s with subscripts are the ones in Theorem 3.1 and the $\bar{\Theta}$'s with subscripts are similar restrictions of $\bar{\Theta}$ with training or test inputs. Furthermore, if $\mathcal{H}$ is the inverse gamma with shape $a$ and scale $b$, the exact marginal distribution of $(f_\infty^\infty(\mathbf{x}_{K+i}))_{i\in[L]}$ is the following multivariate $t$ distribution:*

$$(f_\infty^\infty(\mathbf{x}_{K+i}))_{i\in[L]} \sim \mathrm{MVT}_L(2a, \mu', (b/a) \cdot \Theta'). \tag{5}$$

*Remark* (Unifying High-level Principles). All of our results and their proofs are derived from two high-level principles. First, once we condition on the variance $\sigma_v^2$ of the readout layer's parameters, the networks in our setup fit into the standard setting for NNGPs and NTKs, so that they satisfy the existing results in the setting. This conditioning trick means that most of the results in the standard setting can be carried over to our setup in a scale-mixture form, as we explained in this section. Second, if the prior on $\sigma_v^2$ is inverse gamma, the marginalization of $\sigma_v^2$ in those results can be calculated analytically and have a form of Student's $t$ distribution or process.

### 3.1 POSTERIOR INFERENCE

**Gaussian likelihood**  As for the NNGPs, when the scale is mixed by the inverse gamma prior, we can exactly compute the posterior predictives of the scale mixture of NNGPs for Gaussian likelihood since it corresponds to the Student's $t$ process having closed-form characterizations for the posteriors. For generic prior settings for the scales, we no longer have such closed-form formulas for predictive posteriors, and have to rely on approximate inference methods. We describe one such method, namely, self-normalizing importance sampling with prior as proposal, together with a technique for optimizing the method specifically for scale mixtures of NNGPs.

Consider a scale mixture of NNGPs with a prior $\mathcal{H}$ on the variance $\sigma_v^2$. Assume that we want to estimate the expectation of $h(y)$ for some $h : \mathbb{R} \to \mathbb{R}$ where the random variable $y$ is drawn from the predictive posterior of the mixture at some input $\mathbf{x} \in \mathbb{R}^I$ under the condition on $\mathcal{D}_{\text{tr}}$. The importance sampling with prior as proposal computes the estimator $(\sum_{i=1}^N w_i h(y_i))/\sum_{j=1}^N w_j$ where $\beta_i$ and $y_i$ for $i = 1, \ldots, N$ are independent samples from the prior $\mathcal{H}$ and the posterior of the Gaussian distribution, respectively, and the $w_i$'s are the importance weights of the $\beta_i$'s:

$$\beta_i \sim \mathcal{H}, \quad w_i = \mathcal{N}(Y_{\text{tr}}; 0, \beta_i \overline{\mathcal{K}}_{\text{tr,tr}}), \quad y_i \sim \mathcal{N}(\overline{\mathcal{K}}_{\mathbf{x},\text{tr}} \overline{\mathcal{K}}_{\text{tr,tr}}^{-1} Y_{\text{tr}}, \beta_i(\overline{\mathcal{K}}_{\mathbf{x},\mathbf{x}} - \overline{\mathcal{K}}_{\mathbf{x},\text{tr}} \overline{\mathcal{K}}_{\text{tr,tr}}^{-1} \overline{\mathcal{K}}_{\text{tr,}\mathbf{x}})).$$

The $\overline{\mathcal{K}}$ is the covariance matrix computed as in Theorem 3.1 except that $\mathbf{x}$ is now a test input.

A naive implementation of this importance sampler is slow unnecessarily due to the inefficient calculation of the likelihoods $\mathcal{N}(Y_{\text{tr}}; 0, \beta_i \overline{\mathcal{K}}_{\text{tr,tr}})$ of the sampled $\beta_i$'s. To see this, note that the log likelihood of $\beta_i$ can be decomposed as follows:

$$\log \mathcal{N}(Y_{\text{tr}}; 0, \beta_i \overline{\mathcal{K}}_{\text{tr,tr}}) = -\frac{K}{2} \log(2\pi) - \frac{1}{2} \log \det \left( \overline{\mathcal{K}}_{\text{tr,tr}} \right) - \frac{K}{2} \log(\beta_i) - \frac{1}{2\beta_i} Y_{\text{tr}}^\top \overline{\mathcal{K}}_{\text{tr,tr}}^{-1} Y_{\text{tr}}.$$

The terms $\frac{K}{2} \log(2\pi)$, $\frac{1}{2} \log \det(\overline{\mathcal{K}}_{\text{tr,tr}})$, and $Y_{\text{tr}}^\top \overline{\mathcal{K}}_{\text{tr,tr}}^{-1} Y_{\text{tr}}$ are shared across all the samples $\{\beta_i\}_{i=1}^N$, so need not be computed everytime we draw $\beta_i$. To avoid these duplicated calculations, we compute these shared terms beforehand, so that the log-likliehood for each $\beta_i$ can be computed in $O(1)$ time.

**Generic likelihoods**  For generic likelihoods other than Gaussian, even the vanilla NNGPs do not admit closed-form posterior predictives. In such cases, we can employ the Sparse Variational Gaussian Process (SVGP) (Titsias, 2009) for approximate inference. We present a similar algorithm for the inverse-gamma mixing case which leads to Student's $t$ process. See Appendices C and D for the detailed description.

## 4  EXPERIMENTS

We empirically evaluated the scale mixtures of NNGPs on various synthetic and real-world tasks. We tested the scale mixture of NNGPs with inverse-gamma prior corresponding to Student's $t$ processes, and with another heavy-tailed prior called Burr Type XII distribution (Appendix H). Our implementation used Neural Tangents library (Novak et al., 2020) and JAX (Bradbury et al., 2018).

### 4.1  EMPIRICAL VALIDATION OF THE THEORIES

To empirically validate our theories, we set up a fully connected network having three layers with 512 hidden units and erf activation function. Except for the last layer, the weights of the network were initialized with $\mathcal{N}(0, 8/n)$ and the biases were initialized with $\mathcal{N}(0, 0.05^2/n)$. For the last layer, we sampled $\sigma_v^2 \sim \text{InvGamma}(a, b)$ with varying $(a, b)$ values and sampled weights from $\mathcal{N}(0, \sigma_v^2)$. To check Theorem 3.1, we initialize 1,000 models and computed the distribution of outputs evaluated at zero. For Theorem 3.2, using $y = \sin(x)$ as the target function, we first initialized the parameters, trained only the last layer for multiple $\sigma_v$ values, and averaged the results to get function value at zero. Similarly, we trained all the layers to check Theorem 3.3. As shown in Fig. 1, the theoretical limit well-matched the empirically obtained distributions for all settings we tested. See Appendix G for the details and more results.

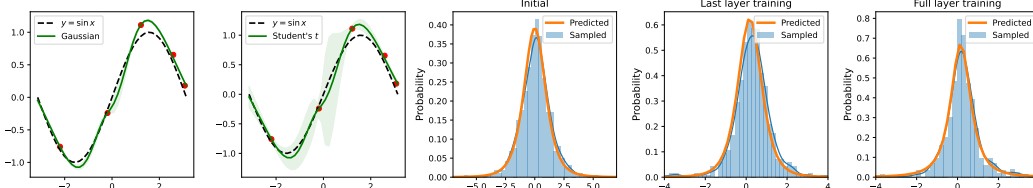

**Figure 1:** (1st-2nd column) Posterior predictives obtained from NNGP (left) and scale mixture of NNGPs with inverse gamma prior (right). (3rd-5th column) correspondence between wide finite model vs theoretical limit.

**Table 1:** NLL values on UCI dataset. $(m, d)$ denotes number of data points and features, respectively. We take results from Adlam et al. (2020) except our model.

| Dataset | $(m, d)$ | PBP-MV | Dropout | Ensembles | RBF | NNGP | Ours |
|---|---|---|---|---|---|---|---|
| Boston Housing | (506, 13) | $2.54 \pm 0.08$ | $\mathbf{2.40} \pm 0.04$ | $2.41 \pm 0.25$ | $2.63 \pm 0.09$ | $2.65 \pm 0.13$ | $2.72 \pm 0.05$ |
| Concrete Strength | (1030, 8) | $3.04 \pm 0.03$ | $\mathbf{2.93} \pm 0.02$ | $3.06 \pm 0.18$ | $3.52 \pm 0.11$ | $3.19 \pm 0.05$ | $3.13 \pm 0.04$ |
| Energy Efficiency | (768, 8) | $1.01 \pm 0.01$ | $1.21 \pm 0.01$ | $1.38 \pm 0.22$ | $0.78 \pm 0.06$ | $1.01 \pm 0.04$ | $\mathbf{0.67} \pm 0.04$ |
| Kin8nm | (8192, 8) | $\mathbf{-1.28} \pm 0.01$ | $-1.14 \pm 0.01$ | $-1.20 \pm 0.02$ | $-1.11 \pm 0.01$ | $-1.15 \pm 0.01$ | $-1.18 \pm 0.01$ |
| Naval Propulsion | (11934, 16) | $-4.85 \pm 0.06$ | $-4.45 \pm 0.00$ | $-5.63 \pm 0.05$ | $\mathbf{-10.07} \pm 0.01$ | $-10.01 \pm 0.01$ | $-8.04 \pm 0.04$ |
| Power Plant | (9568, 4) | $2.78 \pm 0.01$ | $2.80 \pm 0.01$ | $2.79 \pm 0.04$ | $2.94 \pm 0.01$ | $2.77 \pm 0.02$ | $\mathbf{2.66} \pm 0.01$ |
| Wine Quality Red | (1588, 11) | $0.97 \pm 0.01$ | $0.93 \pm 0.01$ | $0.94 \pm 0.12$ | $-0.78 \pm 0.07$ | $\mathbf{-0.98} \pm 0.06$ | $-0.77 \pm 0.07$ |
| Yacht Hydrodynamics | (308, 6) | $1.64 \pm 0.02$ | $1.25 \pm 0.01$ | $1.18 \pm 0.21$ | $0.49 \pm 0.06$ | $1.07 \pm 0.27$ | $\mathbf{0.17} \pm 0.25$ |

## 4.2 REGRESSION WITH GAUSSIAN LIKELIHOODS

We tested the scale mixtures of NNGPs and other models on various real-world regression tasks. All the models are trained with the squared loss function. We expect the scale mixtures of NNGPs to be better calibrated than NNGP due to their heavy-tail behaviors. To see this, we fitted the scale mixture of NNGPs with inverse gamma prior on eight datasets collected from UCI repositories [1] and measured the negative log-likelihood values on the test set. Table 1 summarizes the result. The results other than ours are borrowed from Adlam et al. (2020). In summary, ours generally performed similarly to NNGP except for some datasets for which ours significantly outperformed NNGP. Considering the fact that Student's $t$ processes include GPs as limiting cases, this result suggests that one can use the scale mixtures as an alternative to NNGPs even for the datasets not necessarily including heavy-tailed noises. See Appendix G for detailed settings for the training.

## 4.3 CLASSIFICATION WITH GAUSSIAN LIKELIHOOD

Following the literature, we apply the NNGPs and the scale mixtures of NNGPs to classification problems with Gaussian likelihoods (squared loss). We summarize the results in Appendix F. As a brief summary, ours with heavy-tailed priors (inverse gamma, Burr Type XII) outperformed NNGP in terms of uncertainty calibration for various datasets including corrupted datasets.

## 4.4 CLASSIFICATION WITH CATEGORICAL LIKELIHOOD

We compared the NNGP and the scale mixture of NNGPs with inverse-gamma priors on the image classification task. Following the standard setting in image classification with deep neural networks, we computed the posterior predictive distributions of the BNNs under categorical likelihood. Since both NNGPs and the scale mixtures of NNGPs do not admit closed-form posteriors, we employed SVGP as an approximate inference method for NNGP (Appendix C), and extended the SVGP for the scale mixture of NNGPs (Appendix D). We call the latter as Sparse Variational Student's $t$ Process (SVTP) since it approximates the posteriors whose limit corresponds to Student's $t$ process. We compared SVGP and SVTP on multiple image classification benchmarks, including MNIST, CIFAR10, and SVHN. We also evaluated the predictive performance on Out-Of-Distribution (OOD) data for which we intentionally removed three training classes to save as OOD classes to be tested. We used four-layer CNN as a base model to compute NNGP kernels. See Appendix G for details. Table 2 summarizes the results, which show that SVTP and SVGP perform similarly for in-distribution data,

---

[1] https://archive.ics.uci.edu/ml/datasets.php

**Table 2:** Classification accuracy and NLL of SVGP and SVTP for image datasets and their variants. NLL values are multiplied by $10^2$.

| | SVGP | | SVTP (Ours) | | | SVGP | | SVTP (Ours) | |
|---|---|---|---|---|---|---|---|---|---|
| Dataset | NLL ($\times 10^2$) | Accuracy (%) | NLL ($\times 10^2$) | Accuracy (%) | Dataset | NLL ($\times 10^2$) | Accuracy (%) | NLL ($\times 10^2$) | Accuracy (%) |
| MNIST | $8.96 \pm 0.12$ | $97.73 \pm 0.03$ | $\mathbf{8.90 \pm 0.04}$ | $\mathbf{97.78 \pm 0.04}$ | CIFAR10 | $131.96 \pm 0.35$ | $54.09 \pm 0.12$ | $132.13 \pm 0.24$ | $\mathbf{54.10 \pm 0.13}$ |
| + Shot | $\mathbf{24.22 \pm 0.08}$ | $94.63 \pm 0.05$ | $24.28 \pm 0.10$ | $\mathbf{94.63 \pm 0.07}$ | + Shot 5 | $143.11 \pm 0.29$ | $49.43 \pm 0.14$ | $\mathbf{142.30 \pm 0.18}$ | $\mathbf{49.73 \pm 0.05}$ |
| + Impulse | $\mathbf{56.52 \pm 0.88}$ | $\mathbf{90.29 \pm 0.65}$ | $57.91 \pm 0.57$ | $89.36 \pm 0.58$ | + Impulse 5 | $164.90 \pm 0.11$ | $41.66 \pm 0.19$ | $\mathbf{160.79 \pm 0.75}$ | $\mathbf{43.08 \pm 0.34}$ |
| + Spatter | $16.79 \pm 0.19$ | $95.95 \pm 0.04$ | $\mathbf{16.66 \pm 0.05}$ | $\mathbf{95.99 \pm 0.04}$ | + Spatter 5 | $\mathbf{141.11 \pm 0.30}$ | $\mathbf{50.34 \pm 0.17}$ | $141.14 \pm 0.15$ | $50.31 \pm 0.15$ |
| + Glass Blur | $100.65 \pm 2.35$ | $62.63 \pm 0.65$ | $\mathbf{97.19 \pm 2.23}$ | $\mathbf{63.52 \pm 0.74}$ | + Fog 5 | $213.50 \pm 0.19$ | $25.47 \pm 0.21$ | $\mathbf{209.31 \pm 0.48}$ | $\mathbf{26.03 \pm 0.16}$ |
| w. OOD | $216.58 \pm 1.83$ | $\mathbf{67.70 \pm 0.03}$ | $\mathbf{206.86 \pm 1.85}$ | $67.67 \pm 0.03$ | + Snow 5 | $166.44 \pm 0.51$ | $\mathbf{41.47 \pm 0.27}$ | $\mathbf{166.41 \pm 0.28}$ | $41.47 \pm 0.11$ |
| | | | | | w. OOD | $341.59 \pm 1.75$ | $\mathbf{41.83 \pm 0.11}$ | $\mathbf{333.70 \pm 1.83}$ | $41.19 \pm 0.17$ |
| KMNIST | $\mathbf{53.93 \pm 0.13}$ | $83.92 \pm 0.09$ | $53.95 \pm 0.30$ | $\mathbf{83.96 \pm 0.08}$ | EMNIST | $56.49 \pm 1.24$ | $84.25 \pm 0.22$ | $\mathbf{54.92 \pm 0.84}$ | $\mathbf{84.55 \pm 0.32}$ |
| w. OOD | $268.41 \pm 2.40$ | $\mathbf{60.76 \pm 0.06}$ | $\mathbf{257.16 \pm 1.86}$ | $60.46 \pm 0.05$ | w. OOD | $183.25 \pm 1.40$ | $\mathbf{72.58 \pm 0.28}$ | $\mathbf{177.65 \pm 0.43}$ | $72.38 \pm 0.16$ |
| Fashion MNIST | $34.30 \pm 0.10$ | $87.84 \pm 0.13$ | $\mathbf{34.25 \pm 0.13}$ | $\mathbf{87.90 \pm 0.05}$ | SVHN | $\mathbf{100.88 \pm 0.17}$ | $71.67 \pm 0.11$ | $101.04 \pm 0.13$ | $\mathbf{71.71 \pm 0.06}$ |
| w. OOD | $252.61 \pm 3.51$ | $\mathbf{62.29 \pm 0.04}$ | $\mathbf{241.69 \pm 2.45}$ | $62.24 \pm 0.06$ | w. OOD | $379.75 \pm 9.24$ | $\mathbf{46.86 \pm 0.19}$ | $\mathbf{360.40 \pm 3.68}$ | $46.43 \pm 0.08$ |

**Table 3:** Classification accuracy and NLL of ensemble models for image datasets and their variants. We used 8 models of 4-layer CNN for our base ensemble model. NLL values are multiplied by $10^2$.

| | Gaussian | | Inverse Gamma Prior (Ours) | | | Gaussian | | Inverse Gamma Prior (Ours) | |
|---|---|---|---|---|---|---|---|---|---|
| Dataset | NLL ($\times 10^2$) | Accuracy (%) | NLL ($\times 10^2$) | Accuracy (%) | Dataset | NLL ($\times 10^2$) | Accuracy (%) | NLL ($\times 10^2$) | Accuracy (%) |
| MNIST | $0.33 \pm 0.01$ | $98.94 \pm 0.04$ | $\mathbf{0.32 \pm 0.01}$ | $\mathbf{98.98 \pm 0.04}$ | KMNIST | $2.74 \pm 0.04$ | $93.24 \pm 0.20$ | $\mathbf{2.75 \pm 0.02}$ | $\mathbf{93.34 \pm 0.16}$ |
| + Shot | $\mathbf{1.42 \pm 0.14}$ | $\mathbf{95.73 \pm 0.48}$ | $1.42 \pm 0.17$ | $95.73 \pm 0.46$ | w. OOD | $78.64 \pm 2.14$ | $\mathbf{65.64 \pm 0.07}$ | $\mathbf{70.06 \pm 3.14}$ | $65.61 \pm 0.13$ |
| + Impulse | $\mathbf{5.91 \pm 0.68}$ | $\mathbf{83.10 \pm 1.32}$ | $6.27 \pm 0.78$ | $82.80 \pm 1.31$ | w. Imbalance | $9.47 \pm 0.27$ | $77.04 \pm 0.96$ | $\mathbf{9.12 \pm 0.57}$ | $\mathbf{77.23 \pm 1.67}$ |
| + Spatter | $0.76 \pm 0.02$ | $97.58 \pm 0.12$ | $\mathbf{0.74 \pm 0.02}$ | $\mathbf{97.63 \pm 0.03}$ | w. Noisy Label | $10.78 \pm 0.03$ | $\mathbf{84.42 \pm 0.10}$ | $\mathbf{10.67 \pm 0.03}$ | $84.30 \pm 0.12$ |
| + Glass Blur | $3.25 \pm 0.31$ | $88.50 \pm 1.27$ | $\mathbf{2.83 \pm 0.20}$ | $\mathbf{90.22 \pm 0.80}$ | | | | | |
| w. OOD | $82.71 \pm 2.65$ | $\mathbf{68.51 \pm 0.01}$ | $\mathbf{74.31 \pm 2.93}$ | $68.47 \pm 0.03$ | Fashion MNIST | $\mathbf{2.36 \pm 0.04}$ | $\mathbf{91.92 \pm 0.11}$ | $2.37 \pm 0.02$ | $91.82 \pm 0.11$ |
| w. Imbalance | $\mathbf{1.50 \pm 0.21}$ | $\mathbf{95.85 \pm 0.44}$ | $1.50 \pm 0.23$ | $95.82 \pm 0.49$ | w. OOD | $62.17 \pm 0.50$ | $64.47 \pm 0.09$ | $\mathbf{58.57 \pm 0.97}$ | $\mathbf{64.48 \pm 0.07}$ |
| w. Noisy Label | $7.70 \pm 0.06$ | $\mathbf{97.62 \pm 0.04}$ | $\mathbf{7.63 \pm 0.06}$ | $97.56 \pm 0.03$ | w. Imbalance | $5.54 \pm 0.03$ | $\mathbf{83.09 \pm 0.12}$ | $\mathbf{5.63 \pm 0.08}$ | $83.00 \pm 0.11$ |
| CIFAR10 | $\mathbf{8.68 \pm 0.06}$ | $\mathbf{70.77 \pm 0.31}$ | $8.74 \pm 0.08$ | $70.22 \pm 0.30$ | w. Noisy Label | $9.30 \pm 0.08$ | $87.05 \pm 0.07$ | $\mathbf{9.20 \pm 0.06}$ | $\mathbf{87.27 \pm 0.19}$ |
| + Shot 5 | $16.50 \pm 0.27$ | $51.48 \pm 0.34$ | $\mathbf{15.28 \pm 0.38}$ | $\mathbf{53.28 \pm 0.37}$ | | | | | |
| + Impulse 5 | $32.39 \pm 1.11$ | $33.51 \pm 1.24$ | $\mathbf{29.20 \pm 1.49}$ | $\mathbf{35.66 \pm 1.56}$ | EMNIST | $0.76 \pm 0.01$ | $91.25 \pm 0.13$ | $\mathbf{0.75 \pm 0.01}$ | $\mathbf{91.30 \pm 0.05}$ |
| + Spatter 5 | $12.83 \pm 0.23$ | $58.60 \pm 0.59$ | $\mathbf{12.36 \pm 0.17}$ | $\mathbf{59.36 \pm 0.26}$ | w. OOD | $8.37 \pm 0.15$ | $76.59 \pm 0.09$ | $\mathbf{8.25 \pm 0.17}$ | $\mathbf{76.72 \pm 0.08}$ |
| + Fog 5 | $15.40 \pm 0.07$ | $45.23 \pm 0.07$ | $\mathbf{15.28 \pm 0.06}$ | $\mathbf{45.73 \pm 0.22}$ | w. Noisy Label | $3.23 \pm 0.02$ | $\mathbf{86.64 \pm 0.17}$ | $\mathbf{3.15 \pm 0.01}$ | $86.25 \pm 0.20$ |
| + Snow 5 | $13.18 \pm 0.25$ | $57.08 \pm 0.48$ | $\mathbf{12.76 \pm 0.08}$ | $\mathbf{57.66 \pm 0.37}$ | | | | | |
| w. OOD | $81.57 \pm 0.01$ | $\mathbf{50.66 \pm 0.13}$ | $\mathbf{80.62 \pm 0.86}$ | $50.46 \pm 0.23$ | SVHN | $\mathbf{4.68 \pm 0.05}$ | $\mathbf{87.84 \pm 0.17}$ | $4.70 \pm 0.04$ | $87.82 \pm 0.13$ |
| w. Imbalance | $19.22 \pm 0.12$ | $38.73 \pm 1.35$ | $\mathbf{19.05 \pm 0.18}$ | $\mathbf{39.74 \pm 0.37}$ | w. OOD | $105.17 \pm 1.75$ | $\mathbf{56.92 \pm 0.03}$ | $\mathbf{101.87 \pm 1.92}$ | $56.92 \pm 0.13$ |
| w. Noisy Label | $\mathbf{15.88 \pm 0.03}$ | $\mathbf{54.90 \pm 0.34}$ | $15.90 \pm 0.05$ | $54.81 \pm 0.38$ | w. Imbalance | $\mathbf{14.20 \pm 0.17}$ | $63.64 \pm 0.32$ | $14.88 \pm 0.19$ | $\mathbf{61.97 \pm 0.92}$ |
| | | | | | w. Noisy Label | $\mathbf{12.22 \pm 0.11}$ | $\mathbf{80.24 \pm 0.20}$ | $12.36 \pm 0.07$ | $79.84 \pm 0.24$ |

but SVTP significantly outperforms SVGP for OOD data in terms of uncertainty calibration (measured by negative log-likelihoods).

### 4.5 CLASSIFICATION BY FINITE MODEL ENSEMBLE

Although we proved Theorem 3.3 only for fully-connected networks trained with squared losses, Lee et al. (2019) empirically observed that there still seem to be a similar correspondence for neural networks trained with cross-entropy loss. To this end, we tested the ensembles of wide finite CNNs with 4 layers trained by gradient descent over the cross-entropy loss. Each model in our test is initialized by the NTK parameterization with random scales drawn from inverse gamma prior on the last layer (ours) or without such random scales (NNGP). For each prior setting, we constructed ensembles of 8 models, and compared the performance on MNIST, MNIST-like variants, CIFAR10, and SVHN. We also compared the models under the presence of corruption, OOD data, label imbalance, and label noises. See Appendix G for detailed description for the experimental setting. As for the previous experiments, Table 3 demonstrates that ours with inverse gamma prior largely outperforms the baseline, especially for the OOD or corrupted data.

## 5 CONCLUSION

In this paper, we proposed a simple extension of NNGPs by introducing a scale prior on the last-layer weight parameters. The resulting method, entitled as the scale mixture of NNGPs, defines a broad class of stochastic processes, especially the heavy-tailed ones such as Student's $t$ processes. Based on the result in (Yang, 2019), we have shown that an infinitely-wide BNN of any architecture constructed in a specific way corresponds to the scale mixture of NNGPs. Also, we have extended the existing convergence results of infinitely-wide BNNs trained with gradient descent to GPs so that the results hold for our construction. Our empirical evaluation validates our theory and shows promising results for multiple real-world regression and classification tasks. Especially, it shows that the heavy-tailed processes from our construction are robust to the out-of-distribution data.

## ACKNOWLEDGEMENTS AND DISCLOSURE OF FUNDING

We would like to thank Jaehoon Lee for helping us to understand his results on NNGP and NTK. This work was partly supported by Institute of Information & communications Technology Planning & Evaluation (IITP) grant funded by the Korea government (MSIT) (No.2019-0-00075, Artificial Intelligence Graduate School Program(KAIST), No. 2021-0-02068, Artificial Intelligence Innovation Hub), National Research Foundation of Korea (NRF) funded by the Ministry of Education (NRF2021R1F1A1061655, NRF-2021M3E5D9025030). HY was supported by the Engineering Research Center Program through the National Research Foundation of Korea (NRF) funded by the Korean Government MSIT (NRF-2018R1A5A1059921) and also by the Institute for Basic Science (IBS-R029-C1).

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

## A  DETAILS ON TENSOR PROGRAMS

In this section, we will review tensor programs from (Yang, 2019), which are sequences of assignment statements followed by a return statement where each variable is updated at most once.

Tensor programs use the three types of variables: *G-variables*, *H-variables*, and *A-variables*. G- and H-variables are vector-type variables, and A-variables are matrix-type variables. Each tensor program is parameterized by $n \in \mathbb{N}$, which determines the dimension of all vectors (i.e., G- and H-variables) and matrices (i.e., A-variables) in the program to $n$ and $n \times n$, respectively. Due to the random initialization of some of these variables, the contents of these variables are random in general. G-variables denote Gaussian-distributed random variables, and H-variables store the results of applying (usually nonlinear) functions on G-variables. A-variables store random matrices whose entries are set with iid samples from a Gaussian distribution. All H-variables in a tensor program are set their values by some assignments, but some G-variables and all H-variables may not be assigned to in the program. Such G-variables are called input G-variables.

Assignments in tensor programs have one of the three kinds: *MatMul*, *LinComb*, and *Nonlin*. The MatMul assignments have the form $y = Wx$ where $y$ is a $G$-variable, $W$ is an A-variable, and $x$ is a H-variable. The LinComb assignments compute linear combinations, and they have the form $y = \sum_{i=1}^{l} w_i x^i$ where $y, x^1, \ldots, x^l$ are G-variables, and $w_1, \ldots, w_l \in \mathbb{R}$ are constants. The final Nonlin assignments apply (usually nonlinear) scala functions on the values of G-variables coordinatewise. That is, they have the form $y = \phi(x^1, \ldots, x^l)$, where $y$ is an H-variable, $x^1, \ldots, x^l$ are G-variables, $\phi : \mathbb{R}^l \to \mathbb{R}$ is a possibly nonlinear function, and $\phi(x^1, \ldots, x^l)$ means the lifted application of $\phi$ to vector arguments.

Every tensor program ends with a return statement of the form:

$$\texttt{return}(v^{1\top} y^1 / \sqrt{n}, \ldots, v^{l\top} y^l / \sqrt{n}), \tag{6}$$

where $v^1, \ldots, v^l$ are G-variables not appearing anywhere in the program, and $y^1, \ldots, y^l$ are H-variables.

**Assumption A.1.** *Every A-variable $W$ in a tensor program is initialized with iid samples from the Gaussian distribution with mean zero and covariance $\sigma_W^2/n$ for some $\sigma_W > 0$ (i.e., $W_{ij} \sim \mathcal{N}(0, \sigma_W^2/n)$ for $i, j \in [n]$), and these samples are independent with those used for other A-variables and input G-variables. For every $\alpha \in [n]$, the components of all input G-variables $x^1, \ldots, x^m$ (which store vectors in $\mathbb{R}^n$) are initialized with independent samples from $\mathcal{N}(\mu^{in}, \Sigma^{in})$ for some mean $\mu^{in}$ and (possibly singular) covariance $\Sigma^{in}$ over all input G-variables. Note that the mean and the covariance do not depend on the component index $\alpha \in [n]$. Finally, the input G-variables $v^i$ used in the return statement of the program are independent with all the other random variables in the program, and their components are set by iid samples from $\mathcal{N}(0, \sigma_v^2)$.*

**Definition A.1.** Given a tensor program satisfying Assumption A.1, we can compute $\mu$ and $\Sigma$ for not only input G-variables but all the G-variables in the program. Here the dimension of the vector $\mu$ is the number $N$ of all G-variables. The computed $\mu$ and $\Sigma$ specify the multivariate Gaussian distribution over the $\alpha$-th components of G-variables that arises at the infinite-width limit (Theorem A.2). The computation is defined inductively as follows. Let $g^1, \ldots, g^N$ be all the G-variables in the program, ordered as they appear in the program. For any G-variable $g^k$, $\mu(g^k)$ is defined by:

$$\mu(g^k) = \begin{cases} \mu^{in}(g^k) & \text{if } g^k \text{ is an input G-variable} \\ \sum_{i=1}^{m} w_i \mu(x^i) & \text{if } g^k = \sum_{i=1}^{m} a_i x^i \\ 0 & \text{otherwise} \end{cases}.$$

Here $x^1, \ldots, x^m$ are some G-variables. For any pair of G-variables $g^k, g^l$, $\Sigma(g^k, g^l)$ is defined by:

$$\Sigma(g^k, g^l) = \begin{cases} \Sigma^{in}(g^k, g^l) & \text{if } g^k, g^l \text{ are input G-variables} \\ \sum_{i=1}^{m} w_i \Sigma(x^i, g^l) & \text{if } g^k = \sum_{i=1}^{m} w_i x^i \\ \sum_{i=1}^{m} w_i \Sigma(g^k, x^i) & \text{if } g^l = \sum_{i=1}^{m} w_i x^i \\ \sigma_W^2 \mathbb{E}_Z[\phi(Z)\phi'(Z)] & \text{if } g^k = Wy, g^l = Wy' \text{ for the same } W, \text{ with } Z, \phi, \phi' \text{ below} \\ 0 & \text{otherwise} \end{cases}.$$

Here $x^1, \ldots, x^m$ are some G-variables, and H-variables $y$ and $y'$ in the second last case are introduced by the Nonlin assignments $y = \phi(g^{i_1}, \ldots, g^{i_p})$ and $y' = \phi'(g^{i_1}, \ldots, g^{i_p})$, respectively, for

some $i_1, \ldots, i_p \in [\max(k - 1, l - 1)]$. We have adjusted the inputs of $\phi$ and $\phi'$ to ensure that they take the same set of G-variables. The random variable $Z$ in the second last case is a random Gaussian vector drawn from $\mathcal{N}(\mu', \Sigma')$ where $\mu'$ and $\Sigma'$ are the mean vector and covariance matrix for $g^{i_1}, \ldots, g^{i_p}$ by the previous step of this inductive construction.

**Definition A.2.** Given $h : \mathbb{R}^n \to \mathbb{R}$ and for any $x \in \mathbb{R}^n$ if $|h(x)|$ is upper bounded by a function $\exp(C\|x\|^{2-\epsilon} + c)$ with $C, c, \epsilon > 0$, then this function is *controlled*.

**Theorem A.2** (Master Theorem). *Consider a tensor program satisfying Assumption A.1 and using only controlled $\phi$'s. Let $g^1, \ldots, g^N$ be all of the G-variables that appear in this program, and $h : \mathbb{R}^N \to \mathbb{R}$ be a controlled function. Then, as $n$ tends to $\infty$,*

$$\frac{1}{n} \sum_{\alpha=1}^{n} h(g_\alpha^1, \ldots, g_\alpha^N) \xrightarrow{a.s.} \mathbb{E}_{Z \sim \mathcal{N}(\mu, \Sigma)}[h(Z_1, \ldots, Z_N)].$$

*Here $\xrightarrow{a.s.}$ refers to almost sure convergence, and $Z = (Z_1, \ldots, Z_N) \in \mathbb{R}^N$. Also, $\mu$ and $\Sigma$ are from Definition A.1.*

**Corollary A.3.** *Consider a tensor program satisfying Assumption A.1 and using only controlled $\phi$'s. Let $g^1, \ldots, g^N$ be all of the G-variables that appear in this program. Assume that the program returns a vector $(v^\top y^1 / \sqrt{n}, \ldots, v^\top y^k / \sqrt{n})$ for some H-variables $y^i$'s such that each $y_i$ is updated by $y^i = \phi^i(g^1, \ldots, g^N)$ for a controlled $\phi^i$. Then, as $n$ approaches to $\infty$, the joint distribution of this vector converges weakly to the multivariate Gaussian distribution $\mathcal{N}(0, \mathcal{K})$ with the following covariance matrix:*

$$\mathcal{K}_{(i,j)} = \sigma_v^2 \cdot \mathbb{E}_{Z \sim \mathcal{N}(\mu, \Sigma)}[\phi^i(Z)\phi^j(Z)] \text{ for all } i, j \in [k],$$

*where $\mu$ and $\Sigma$ are from Definition A.1.*

# B  PROOFS

In this section, we provide the proofs of our three main results. We first describe a lemma related to a basic property of Student's $t$ distribution:

**Lemma B.1.** *The class of multivariate $t$ distributions is closed under marginalization and conditioning. That is, when*

$$\mathbf{x} = \begin{bmatrix} \mathbf{x}_1 \\ \mathbf{x}_2 \end{bmatrix} \in \mathbb{R}^{d_1 + d_2} \sim \mathrm{MVT}_{d_1 + d_2}\left(\nu, \begin{bmatrix} \mu_1 \\ \mu_2 \end{bmatrix}, \begin{bmatrix} \Sigma_{11} & \Sigma_{12} \\ \Sigma_{21} & \Sigma_{22} \end{bmatrix}\right), \tag{7}$$

*we have that*

$$\mathbf{x}_1 \sim \mathrm{MVT}_{d_1}(\nu, \mu_1, \Sigma_{11}) \quad \text{and} \quad \mathbf{x}_2 | \mathbf{x}_1 \sim \mathrm{MVT}_{d_2}\left(\nu + d_1, \mu_2', \frac{\nu + c}{\nu + d_1}\Sigma_{22}'\right) \tag{8}$$

*where*

$$\mu_2' = \mu_2 + \Sigma_{21}\Sigma_{11}^{-1}(\mathbf{x}_1 - \mu_1), \quad c = (\mathbf{x}_1 - \mu_1)^\top \Sigma_{11}^{-1}(\mathbf{x}_1 - \mu_1), \quad \Sigma_{22}' = \Sigma_{22} - \Sigma_{21}\Sigma_{11}^{-1}\Sigma_{12}.$$

**Definition B.1.** A distribution on $\mathbb{R}_+$ is a *inverse gamma distribution* with shape parameter $a \in \mathbb{R}_+$ and scale parameter $b \in \mathbb{R}_+$ if it has the following density:

$$p(x) = \frac{b^a}{G(a)}\left(\frac{1}{x}\right)^{a+1} e^{-b/x}$$

where $G(\cdot)$ is the gamma function. To express a random variable drawn from this distribution, we write $x \sim \mathrm{InvGamma}(a, b)$.

**Lemma B.2.** *Let $\Sigma \in \mathbb{R}^{d \times d}$ be a postive definite matrix. Consider $\mu \in \mathbb{R}^n$ and $a, b \in \mathbb{R}_+$. If*

$$\sigma^2 \sim \mathrm{InvGamma}(a, b), \qquad\qquad \mathbf{x} | \sigma^2 \sim \mathcal{N}(\mu, \sigma^2 \Sigma),$$

*then the marginal distribution of $\mathbf{x}$ is $\mathrm{MVT}_d(2a, \mu, \frac{b}{a}\Sigma)$.*

Now we proceed to the proofs of main theorems.

**Theorem 3.1** (Convergence). *As $n$ tends to $\infty$, the random variable $(f_n(\mathbf{x}_i; \mathbf{v}_n, \mathbf{\Psi}_n))_{i \in [K+L]}$ converges in distribution to the following random variable $(f_\infty(\mathbf{x}_i))_{i \in [K+L]}$:*

$$\sigma_v^2 \sim \mathcal{H}, \qquad\qquad (f_\infty(\mathbf{x}_i))_{i \in [K+L]} | \sigma_v^2 \sim \mathcal{N}(0, \sigma_v^2 \cdot \overline{\mathcal{K}}),$$

*where $\overline{\mathcal{K}}_{(i,j)} = \mathbb{E}_{Z \sim \mathcal{N}(\mu, \Sigma)}[\phi(Z_i)\phi(Z_j)]$ for $i, j \in [K+L]$. Furthermore, if $\mathcal{H}$ is the inverse-gamma distribution with shape $a$ and scale $b$, the marginal distribution of $(f_\infty(\mathbf{x}_i))_{i \in [K+L]}$ is the following multivariate $t$ distribution:*

$$(f_\infty(\mathbf{x}_i))_{i \in [K+L]} \sim \mathrm{MVT}_{K+L}(2a, 0, (b/a) \cdot \overline{\mathcal{K}}),$$

*where $\mathrm{MVT}_d(\nu', \mu', \mathcal{K}')$ denotes the $d$-dimensional Student's $t$ distribution with degree of freedom $\nu'$, location $\mu'$, and scale matrix $\mathcal{K}'$.*

*Proof.* Let $o^i[n] = f_n(\mathbf{x}_i; \mathbf{v}_n, \mathbf{\Psi}_n)$ for $i \in [K+L]$ and $n$. Pick an arbitrary bounded continuous function $h : \mathbb{R}^{K+L} \to \mathbb{R}$. For each $n$, define $\mathcal{G}_n$ to be the $\sigma$-field generated by $g_n(\mathbf{x}_1; \mathbf{\Psi}_n), \ldots, g_n(\mathbf{x}_{K+L}; \mathbf{\Psi}_n)$.

We claim that as $n$ tends to $\infty$,

$$\mathbb{E}\left[h\left(o^1[n], \ldots, o^{K+L}[n]\right)\right] \to \mathbb{E}_{\sigma_v^2 \sim \mathcal{H}}\left[\mathbb{E}_{Y^{1:K+L} \sim \mathcal{N}(0, \sigma_v^2 \overline{\mathcal{K}})}\left[h\left(Y^1, \ldots, Y^{K+L}\right)\right]\right].$$

The theorem follows immediately from this claim. The rest of the proof is about showing the claim.

Note that

$$\mathbb{E}\left[h\left(o^1[n], \ldots, o^{K+L}[n]\right)\right] = \mathbb{E}\left[\mathbb{E}\left[h\left(\frac{1}{\sqrt{n}}\mathbf{v}_n^\top \phi^{1:K+L}\right)\middle| \mathcal{G}, \sigma_v^2\right]\right],$$

where $\phi^{1:K+L} = [\phi(g_n(\mathbf{x}_1; \mathbf{\Psi}_n)), \ldots, \phi(g_n(\mathbf{x}_{K+L}; \mathbf{\Psi}_n))] \in \mathbb{R}^{n \times (K+L)}$. Conditioned on $\mathcal{G}$ and $\sigma_v^2$, the random variable $\frac{1}{\sqrt{n}}\mathbf{v}_n^\top \phi^{1:K+L}$ in the nested expectation from above is distributed by a multivariate Gaussian. The mean of this Gaussian distribution is:

$$\mathbb{E}\left[\frac{1}{\sqrt{n}}\mathbf{v}_n^\top \phi^{1:K+L}\middle| \mathcal{G}, \sigma_v^2\right] = \mathbb{E}\left[\mathbf{v}_n^\top \middle| \mathcal{G}, \sigma_v^2\right] \times \frac{1}{\sqrt{n}}\phi^{1:K+L}$$

$$= 0 \in \mathbb{R}^{K+L}.$$

The covariance is:

$$\overline{\mathcal{K}}[n]_{(i,j)} = \mathbb{E}\left[\left(\frac{1}{\sqrt{n}}\mathbf{v}_n^\top \phi(g_n(\mathbf{x}_i; \mathbf{\Psi}_n))\right)\left(\frac{1}{\sqrt{n}}\mathbf{v}_n^\top \phi(g_n(\mathbf{x}_j; \mathbf{\Psi}_n))\right)\middle| \mathcal{G}, \sigma_v^2\right]$$

$$= \frac{1}{n}\sum_{\alpha=1}^{n}\sum_{\beta=1}^{n}\mathbb{E}\left[\mathbf{v}_{n,\alpha}\mathbf{v}_{n,\beta}\middle| \mathcal{G}, \sigma_v^2\right] \times (\phi(g_n(\mathbf{x}_i; \mathbf{\Psi}_n)))_\alpha (\phi(g_n(\mathbf{x}_j; \mathbf{\Psi}_n)))_\beta$$

$$= \frac{1}{n}\sigma_v^2 \sum_{\alpha=1}^{n}(\phi(g_n(\mathbf{x}_i; \mathbf{\Psi}_n)))_\alpha (\phi(g_n(\mathbf{x}_j; \mathbf{\Psi}_n)))_\alpha.$$

Using what we have observed so far, we compute the limit in our claimed convergence:

$$\lim_{n \to \infty} \mathbb{E}\left[h\left(o^1[n], \ldots, o^{K+L}[n]\right)\right] = \lim_{n \to \infty} \mathbb{E}\left[\mathbb{E}_{u^{1:K+L} \sim \mathcal{N}(0, \overline{\mathcal{K}}[n])}\left[h\left(u^{1:K+L}\right)\right]\right]$$

$$= \mathbb{E}\left[\lim_{n \to \infty} \mathbb{E}_{u^{1:K+L} \sim \mathcal{N}(0, \overline{\mathcal{K}}[n])}\left[h(u^{1:K+L})\right]\right].$$

The last equality follows from the dominated convergence theorem and the fact that $h$ is bounded. To go further in our computation, we observe that $\overline{\mathcal{K}}[n]$ converges almost surely to $\sigma_v^2 \overline{\mathcal{K}}$ by Theorem 2.1 and so

$$\mathbb{E}_{u^{1:K+L} \sim \mathcal{N}(0, \overline{\mathcal{K}}[n])}[h(u^{1:K+L})] = \int h(u^{1:K+L})\mathcal{N}(u^{1:K+L}; o, \overline{\mathcal{K}}[n])\mathrm{d}u^{1:K+L}$$

$$\xrightarrow{a.s.} \int h(u^{1:K+L})\mathcal{N}(u^{1:K+L}; 0, \sigma_v^2 \overline{\mathcal{K}})\mathrm{d}u^{1:K+L}$$

$$= \mathbb{E}_{Y^{1:K+L} \sim \mathcal{N}(0, \sigma_v^2 \overline{\mathcal{K}})}\left[h\left(Y^{1:K+L}\right)\right].$$

Here we use $\mathcal{N}$ not just for the multivariate Gaussian distribution but also for its density, and derive the almost sure convergence from the dominated convergence theorem. Using this observation, we complete our computation:

$$\mathbb{E}\left[\lim_{n\to\infty}\mathbb{E}_{u^{1:K+L}\sim\mathcal{N}(0,\overline{\mathcal{K}}[n])}\left[h(u^{1:K+L})\right]\right]=\mathbb{E}\left[\mathbb{E}_{Y^{1:K+L}\sim\mathcal{N}(0,\sigma_v^2\overline{\mathcal{K}})}\left[h\left(Y^{1:K+L}\right)\right]\right]$$

$$=\mathbb{E}_{\sigma_v^2\sim\mathcal{H}}\left[\mathbb{E}_{Y^{1:K+L}\sim\mathcal{N}(0,\sigma_v^2\overline{\mathcal{K}})}\left[h\left(Y^{1:K+L}\right)\right]\right].$$

$\square$

**Theorem 3.2** (Convergence under Readout-Layer Training). *Assume that we train only the readout layers of the* BNNs *using the training set* $\mathcal{D}_{tr}$ *under mean squared loss and infinitesimal step size. Formally, this means the network parameters are evolved under the following differential equations:*

$$(\mathbf{v}_n^{(0)},\mathbf{\Psi}_n^{(0)})=(\mathbf{v}_n,\mathbf{\Psi}_n),\quad\frac{d\mathbf{v}_n^{(t)}}{dt}=\sum_{i=1}^K\left(y_i-f_n(\mathbf{x}_i;\mathbf{v}_n^{(t)},\mathbf{\Psi}_n^{(t)})\right)\frac{2\phi(g_n(\mathbf{x}_i;\mathbf{\Psi}_n^{(t)}))}{K\sqrt{n}},\quad\frac{d\mathbf{\Psi}_n^{(t)}}{dt}=0.$$

*Then, the time and width limit of the random variables* $(f_n(\mathbf{x}_{K+i};\mathbf{v}_n^{(t)},\mathbf{\Psi}_n^{(t)}))_{i\in[L]}$, *denoted* $(f_\infty^\infty(\mathbf{x}_{K+i}))_{i\in[L]}$, *is distributed by the following mixture of Gaussians:*

$$\sigma_v^2\sim\mathcal{H},\qquad(f_\infty^\infty(\mathbf{x}_{K+i}))_{i\in[L]}|\sigma_v^2\sim\mathcal{N}\left(\left(\overline{\mathcal{K}}_{te,tr}\overline{\mathcal{K}}_{tr,tr}^{-1}Y_{tr}\right),\sigma_v^2\left(\overline{\mathcal{K}}_{te,te}-\overline{\mathcal{K}}_{te,tr}\overline{\mathcal{K}}_{tr,tr}^{-1}\overline{\mathcal{K}}_{tr,te}\right)\right),$$

*where* $Y_{tr}$ *consists of the y values in* $\mathcal{D}_{tr}$ *(i.e.,* $Y_{tr}=(y_1,\dots,y_K)$) *and the* $\overline{\mathcal{K}}$*'s with different subscripts are the restrictions of* $\overline{\mathcal{K}}$ *in Theorem 3.1 with training or test inputs as directed by those subscripts. Furthermore, if* $\mathcal{H}$ *is the inverse-gamma distribution with shape a and scale b, the marginal distribution of* $(f_\infty^\infty(\mathbf{x}_{K+i}))_{i\in[L]}$ *is the following multivariate t distribution:*

$$(f_\infty^\infty(\mathbf{x}_{K+i}))_{i\in[L]}\sim\mathrm{MVT}_L\left(2a,\left(\overline{\mathcal{K}}_{te,tr}\overline{\mathcal{K}}_{tr,tr}^{-1}Y_{tr}\right),\frac{b}{a}\left(\overline{\mathcal{K}}_{te,te}-\overline{\mathcal{K}}_{te,tr}\overline{\mathcal{K}}_{tr,tr}^{-1}\overline{\mathcal{K}}_{tr,te}\right)\right).$$

Our proof of this theorem reuses the general structure of the proof of the corresponding theorem for NNGPs in (Lee et al., 2019), but it adjusts the structure slightly so as to use the master theorem for tensor programs and cover general architectures expressed by those programs. Another new important part of the proof is to condition on the random scale $\sigma_v^2$, so that the existing results such as the master theorem can be used in our case.

*Proof.* Note that $\mathbf{\Psi}_n^{(t)}=\mathbf{\Psi}_n^{(0)}$ for all $t$ because $d\mathbf{\Psi}_n^{(t)}/dt=0$. Let

$$\bar{\phi}_i[n]=\frac{1}{\sqrt{n}}\phi\left(g_n\left(\mathbf{x}_i;\mathbf{\Psi}_n^{(t)}\right)\right)=\frac{1}{\sqrt{n}}\phi\left(g_n\left(\mathbf{x}_i;\mathbf{\Psi}_n^{(0)}\right)\right)\quad\text{for }i\in[K+L],$$

$$\bar{\mathbf{\Phi}}_{tr}[n]=\left[\bar{\phi}_1[n],\dots,\bar{\phi}_K[n]\right]^\top\in\mathbb{R}^{K\times n},\qquad\bar{\mathbf{\Phi}}_{te}[n]=\left[\bar{\phi}_{K+1}[n],\dots,\bar{\phi}_{K+L}[n]\right]^\top\in\mathbb{R}^{L\times n}.$$

Using this notation, we rewrite the differential equation for $\mathbf{v}_n^{(t)}$ as follows:

$$\frac{d\bar{\mathbf{v}}_n^{(t)}}{dt}=-\frac{2}{K}\bar{\mathbf{\Phi}}_{tr}[n]^\top\left(\bar{\mathbf{\Phi}}_{tr}[n]\bar{\mathbf{v}}_n^{(t)}-Y_{tr}\right).$$

This ordinary differential equation has a closed-form solution, which we write below:

$$\bar{\mathbf{v}}_n^{(t)}=\bar{\mathbf{v}}_n^{(0)}+\bar{\mathbf{\Phi}}_{tr}[n]^\top\left(\bar{\mathbf{\Phi}}_{tr}[n]\bar{\mathbf{\Phi}}_{tr}[n]^\top\right)^{-1}\left(\exp\left(-\frac{2}{K}t\bar{\mathbf{\Phi}}_{tr}[n]\bar{\mathbf{\Phi}}_{tr}[n]^\top\right)-I\right)\left(\bar{\mathbf{\Phi}}_{tr}[n]\bar{\mathbf{v}}_n^{(0)}-Y_{tr}\right).$$

We multiply both sides of the equation with the $\bar{\mathbf{\Phi}}_{te}[n]$ matrix, and get

$$\bar{\mathbf{\Phi}}_{te}[n]\bar{\mathbf{v}}_n^{(t)}=\bar{\mathbf{\Phi}}_{te}[n]\bar{\mathbf{v}}_n^{(0)}+\overline{\mathcal{K}}[n]_{te,tr}\overline{\mathcal{K}}[n]_{tr,tr}^{-1}\left(\exp\left(-\frac{2}{K}t\overline{\mathcal{K}}[n]_{tr,tr}\right)-I\right)\left(\bar{\mathbf{\Phi}}_{tr}[n]\bar{\mathbf{v}}_n^{(0)}-Y_{tr}\right)$$

where $\overline{\mathcal{K}}[n]_{te,tr}=\bar{\mathbf{\Phi}}_{te}[n]\bar{\mathbf{\Phi}}_{tr}[n]^\top$ and $\overline{\mathcal{K}}[n]_{tr,tr}=\bar{\mathbf{\Phi}}_{tr}[n]\bar{\mathbf{\Phi}}_{tr}[n]^\top$. Then, we send $t$ to $\infty$, and obtain

$$\bar{\mathbf{\Phi}}_{te}[n]\bar{\mathbf{v}}_n^{(\infty)}=\bar{\mathbf{\Phi}}_{te}[n]\bar{\mathbf{v}}_n^{(0)}-\overline{\mathcal{K}}[n]_{te,tr}\overline{\mathcal{K}}[n]_{tr,tr}^{-1}\left(\bar{\mathbf{\Phi}}_{tr}[n]\bar{\mathbf{v}}_n^{(0)}-Y_{tr}\right)$$

$$=\overline{\mathcal{K}}[n]_{te,tr}\overline{\mathcal{K}}[n]_{tr,tr}^{-1}Y_{tr}+\bar{\mathbf{\Phi}}_{te}[n]\bar{\mathbf{v}}_n^{(0)}-\overline{\mathcal{K}}[n]_{te,tr}\overline{\mathcal{K}}[n]_{tr,tr}^{-1}\left(\bar{\mathbf{\Phi}}_{tr}[n]\bar{\mathbf{v}}_n^{(0)}\right).$$

But

$$(f_n(\mathbf{x}_{K+i}; \mathbf{v}_n^{(\infty)}, \mathbf{\Psi}_n^{(\infty)}))_{i \in [L]} = \bar{\mathbf{\Phi}}_{\text{te}}[n]\, \bar{\mathbf{v}}_n^{(\infty)}.$$

Thus, we can complete the proof of the theorem if we show that for every bounded continuous function $h : \mathbb{R}^L \to \mathbb{R}$,

$$\lim_{n \to \infty} \mathbb{E}\left[ h\left( \overline{\mathcal{K}}[n]_{\text{te,tr}} \overline{\mathcal{K}}[n]_{\text{tr,tr}}^{-1} Y_{\text{tr}} + \bar{\mathbf{\Phi}}_{\text{te}}[n]\, \bar{\mathbf{v}}_n^{(0)} - \overline{\mathcal{K}}[n]_{\text{te,tr}} \overline{\mathcal{K}}[n]_{\text{tr,tr}}^{-1} \left( \bar{\mathbf{\Phi}}_{\text{tr}}[n]\, \bar{\mathbf{v}}_n^{(0)} \right) \right) \right]$$
$$= \mathbb{E}\left[ h\left( (f_\infty^\infty(\mathbf{x}_{K+i}))_{i \in [L]} \right) \right].$$

Pick a bounded continuous function $h : \mathbb{R}^L \to \mathbb{R}$. For each $n$, let $\mathcal{G}_n$ be the $\sigma$-field generated by $g_n(\mathbf{x}_1; \mathbf{\Psi}_n), \ldots, g_n(\mathbf{x}_{K+L}; \mathbf{\Psi}_n)$. Also, define $\overline{\mathcal{K}}[n]_{\text{tr,te}} = \bar{\mathbf{\Phi}}_{\text{tr}}[n] \bar{\mathbf{\Phi}}_{\text{te}}[n]^\top$. We show the sufficient condition mentioned above:

$$\lim_{n \to \infty} \mathbb{E}\left[ h\left( \overline{\mathcal{K}}[n]_{\text{te,tr}} \overline{\mathcal{K}}[n]_{\text{tr,tr}}^{-1} Y_{\text{tr}} + \bar{\mathbf{\Phi}}_{\text{te}}[n]\, \bar{\mathbf{v}}_n^{(0)} - \overline{\mathcal{K}}[n]_{\text{te,tr}} \overline{\mathcal{K}}[n]_{\text{tr,tr}}^{-1} \left( \bar{\mathbf{\Phi}}_{\text{tr}}[n]\, \bar{\mathbf{v}}_n^{(0)} \right) \right) \right]$$

$$= \lim_{n \to \infty} \mathbb{E}\left[ \mathbb{E}\left[ h\left( \overline{\mathcal{K}}[n]_{\text{te,tr}} \overline{\mathcal{K}}[n]_{\text{tr,tr}}^{-1} Y_{\text{tr}} + \bar{\mathbf{\Phi}}_{\text{te}}[n]\, \bar{\mathbf{v}}_n^{(0)} - \overline{\mathcal{K}}[n]_{\text{te,tr}} \overline{\mathcal{K}}[n]_{\text{tr,tr}}^{-1} \left( \bar{\mathbf{\Phi}}_{\text{tr}}[n]\, \bar{\mathbf{v}}_n^{(0)} \right) \right) \,\middle|\, \mathcal{G}_n, \sigma_v^2 \right] \right]$$

$$= \lim_{n \to \infty} \mathbb{E}\left[ \mathbb{E}_{Z \sim \mathcal{N}(\overline{\mathcal{K}}[n]_{\text{te,tr}} \overline{\mathcal{K}}[n]_{\text{tr,tr}}^{-1} Y_{\text{tr}}, \, \sigma_v^2(\overline{\mathcal{K}}[n]_{\text{te,te}} - \overline{\mathcal{K}}[n]_{\text{te,tr}} \overline{\mathcal{K}}[n]_{\text{tr,tr}}^{-1} \overline{\mathcal{K}}[n]_{\text{tr,te}}))} [h(Z)] \right]$$

$$= \mathbb{E}\left[ \lim_{n \to \infty} \mathbb{E}_{Z \sim \mathcal{N}(\overline{\mathcal{K}}[n]_{\text{te,tr}} \overline{\mathcal{K}}[n]_{\text{tr,tr}}^{-1} Y_{\text{tr}}, \, \sigma_v^2(\overline{\mathcal{K}}[n]_{\text{te,te}} - \overline{\mathcal{K}}[n]_{\text{te,tr}} \overline{\mathcal{K}}[n]_{\text{tr,tr}}^{-1} \overline{\mathcal{K}}[n]_{\text{tr,te}}))} [h(Z)] \right]$$

$$= \mathbb{E}\left[ \mathbb{E}_{Z \sim \mathcal{N}(\overline{\mathcal{K}}_{\text{te,tr}} \overline{\mathcal{K}}_{\text{tr,tr}}^{-1} Y_{\text{tr}}, \, \sigma_v^2(\overline{\mathcal{K}}_{\text{te,te}} - \overline{\mathcal{K}}_{\text{te,tr}} \overline{\mathcal{K}}_{\text{tr,tr}}^{-1} \overline{\mathcal{K}}_{\text{tr,te}}))} [h(Z)] \right]$$

$$= \mathbb{E}\left[ h\left( (f_\infty^\infty(\mathbf{x}_{K+i}))_{i \in [L]} \right) \right].$$

The second equality uses the fact that $\mathbf{v}_n^{(0)}$ consists of iid samples from $\mathcal{N}(0, \sigma_v^2)$, and it is a consequence of straightforward calculation. The third equality uses the dominated convergence theorem and the fact that $h$ is bounded. The fourth equality uses the master theorem for tensor programs (Theorem 2.1), the boundedness of $h$, and the dominated convergence theorem. The last equality follows from the definition of $(f_\infty^\infty(\mathbf{x}_{K+i}))_{i \in [L]}$.

If $\sigma_v^2 \sim \text{InvGamma}(a, b)$, we get the following closed-form marginal distribution by Lemma B.2:

$$(f_\infty^\infty(\mathbf{x}_{K+i}))_{i \in [L]} \sim \text{MVT}_L \left( 2a, \, \overline{\mathcal{K}}_{\text{te,tr}} \overline{\mathcal{K}}_{\text{tr,tr}}^{-1} Y_{\text{tr}}, \, \frac{b}{a}(\overline{\mathcal{K}}_{\text{te,te}} - \overline{\mathcal{K}}_{\text{te,tr}} \overline{\mathcal{K}}_{\text{tr,tr}}^{-1} \overline{\mathcal{K}}_{\text{tr,te}}) \right).$$

$\square$

Our proof of Theorem 3.3 uses the corresponding theorem for the standard NTK setup in Lee et al. (2019). We restate this existing theorem in our setup using our notation:

**Theorem B.3.** *Assume that the BNNs are multi-layer perceptrons, and we train these models using the training set $\mathcal{D}_{tr}$ under the mean squared loss and infinitesimal step size. Regard $\sigma_v^2$ as a deterministic value. Then as $n \to \infty$ and $t \to \infty$, $(f_n(\mathbf{x}_{K+i}; \mathbf{v}_n^{(t)}, \frac{1}{\sqrt{n}} \mathbf{\Psi}_n^{(t)}))_{i \in [L]}$ converges in distribution to the following random variable:*

$$(f_\infty^\infty(\mathbf{x}_{K+i}))_{i \in [L]} \sim \mathcal{N}(\mu_{te}^\infty, \Sigma_{te}^\infty)$$

*where*

$$\mu_{te}^\infty = \Theta_{te,tr} \Theta_{tr,tr}^{-1} Y_{tr},$$
$$\Sigma_{te}^\infty = \mathcal{K}_{te,te} + \Theta_{te,tr} \Theta_{tr,tr}^{-1} \mathcal{K}_{tr,tr} \Theta_{tr,tr}^{-1} \Theta_{tr,te} - (\Theta_{te,tr} \Theta_{tr,tr}^{-1} \mathcal{K}_{tr,te} + \mathcal{K}_{te,tr} \Theta_{tr,tr}^{-1} \Theta_{tr,te}).$$

**Theorem 3.3** (Convergence under General Training)**.** *Assume that the BNNs are multi-layer perceptrons and we train all of their parameters using the training set $\mathcal{D}_{tr}$ under mean squared loss and*

*infinitesimal step size. Formally, this means the network parameters are evolved under the following differential equations:*

$$\mathbf{v}_n^{(0)} = \mathbf{v}_n, \qquad \frac{d\mathbf{v}_n^{(t)}}{dt} = \frac{2}{K\sqrt{n}} \sum_{i=1}^{K} \left( y_i - f_n(\mathbf{x}_i; \mathbf{v}_n^{(t)}, \frac{1}{\sqrt{n}} \mathbf{\Psi}_n^{(t)}) \right) \phi(g_n(\mathbf{x}_i; \frac{1}{\sqrt{n}} \mathbf{\Psi}_n^{(t)})),$$

$$\mathbf{\Psi}_n^{(0)} = \mathbf{\Psi}_n, \qquad \frac{d\mathbf{\Psi}_n^{(t)}}{dt} = \frac{2}{K} \sum_{i=1}^{K} \left( y_i - f_n(\mathbf{x}_i; \mathbf{v}_n^{(t)}, \frac{1}{\sqrt{n}} \mathbf{\Psi}_n^{(t)}) \right) \nabla_{\mathbf{\Psi}} f_n(\mathbf{x}_i; \mathbf{v}_n^{(t)}, \frac{1}{\sqrt{n}} \mathbf{\Psi}_n^{(t)}).$$

*Let $\bar{\Theta}$ be $\Theta$ with $\sigma_v^2$ in it set to 1 (so that $\Theta = \sigma_v^2 \bar{\Theta}$). Then, the time and width limit of the random variables $(f_n(\mathbf{x}_{K+i}; \mathbf{v}_n^{(t)}, \frac{1}{\sqrt{n}} \mathbf{\Psi}_n^{(t)}))_{i\in[L]}$ has the following distribution:*

$$\sigma_v^2 \sim \mathcal{H}, \qquad\qquad (f_\infty^\infty(\mathbf{x}_{K+i}))_{i\in[L]} | \sigma_v^2 \sim \mathcal{N}(\mu', \sigma_v^2 \Theta')$$

*where*

$$\mu' = \bar{\Theta}_{te,tr} \bar{\Theta}_{tr,tr}^{-1} Y_{tr},$$

$$\Theta' = \overline{\mathcal{K}}_{te,te} + \left( \bar{\Theta}_{te,tr} \bar{\Theta}_{tr,tr}^{-1} \overline{\mathcal{K}}_{tr,tr} \bar{\Theta}_{tr,tr}^{-1} \bar{\Theta}_{tr,te} \right) - \left( \bar{\Theta}_{te,tr} \bar{\Theta}_{tr,tr}^{-1} \overline{\mathcal{K}}_{tr,te} + \overline{\mathcal{K}}_{te,tr} \bar{\Theta}_{tr,tr}^{-1} \bar{\Theta}_{tr,te} \right).$$

*Here the $\overline{\mathcal{K}}$'s with subscripts are the ones in Theorem 3.1 and the $\bar{\Theta}$'s with subscripts are similar restrictions of $\bar{\Theta}$ with training or test inputs. Furthermore, if $\mathcal{H}$ is the inverse gamma with shape $a$ and scale $b$, the exact marginal distribution of $(f_\infty^\infty(\mathbf{x}_{K+i}))_{i\in[L]}$ is the following multivariate $t$ distribution:*

$$(f_\infty^\infty(\mathbf{x}_{K+i}))_{i\in[L]} \sim \mathrm{MVT}_L(2a, \mu', (b/a) \cdot \Theta'). \tag{5}$$

*Proof.* Assume that we condition on $\sigma_v^2$. Then, by Theorem B.3, as we send $n$ to $\infty$ and then $t$ to $\infty$, the conditioned random variable $(f_n(\mathbf{x}_{K+i}; \mathbf{v}_n^{(t)}, \frac{1}{\sqrt{n}} \mathbf{\Psi}_n^{(t)}))_{i\in[L]} | \sigma_v^2$ converges in distribution to the following random variable:

$$(f_\infty^\infty(\mathbf{x}_{K+i}))_{i\in[L]} | \sigma_v^2 \sim \mathcal{N}(\mu_{te}^\infty, \Sigma_{te}^\infty),$$

where

$$\mu_{te}^\infty = \bar{\Theta}_{te,tr} \bar{\Theta}_{tr,tr}^{-1} Y_{te},$$

$$\Sigma_{te}^\infty = \sigma_v^2 \left( \overline{\mathcal{K}}_{te,te} + \bar{\Theta}_{te,tr} \bar{\Theta}_{tr,tr}^{-1} \overline{\mathcal{K}}_{tr,tr} \bar{\Theta}_{tr,tr}^{-1} \bar{\Theta}_{tr,te} - (\bar{\Theta}_{te,tr} \bar{\Theta}_{tr,tr}^{-1} \overline{\mathcal{K}}_{tr,te} + \overline{\mathcal{K}}_{te,tr} \bar{\Theta}_{tr,tr}^{-1} \bar{\Theta}_{tr,te}) \right).$$

Thus, by Lemma B.4, if we remove the conditioning on $\sigma_v^2$ (i.e., we marginalize over $\sigma_v^2$), we get the convergence of the unconditioned $(f_n(\mathbf{x}_{K+i}; \mathbf{v}_n^{(t)}, \frac{1}{\sqrt{n}} \mathbf{\Psi}_n^{(t)}))_{i\in[L]}$ to the following random variable:

$$\sigma_v^2 \sim \mathcal{H}, \qquad\qquad (f_\infty^\infty(\mathbf{x}_{K+i}))_{i\in[L]} \sim \mathcal{N}(\mu', \sigma_v^2 \Theta')$$

where

$$\mu' = \bar{\Theta}_{te,tr} \bar{\Theta}_{tr,tr}^{-1} Y_{tr},$$

$$\Theta' = \overline{\mathcal{K}}_{te,te} + \left( \bar{\Theta}_{te,tr} \bar{\Theta}_{tr,tr}^{-1} \overline{\mathcal{K}}_{tr,tr} \bar{\Theta}_{tr,tr}^{-1} \bar{\Theta}_{tr,te} \right) - \left( \bar{\Theta}_{te,tr} \bar{\Theta}_{tr,tr}^{-1} \overline{\mathcal{K}}_{tr,te} + \overline{\mathcal{K}}_{te,tr} \bar{\Theta}_{tr,tr}^{-1} \bar{\Theta}_{tr,te} \right).$$

In particular, when $\mathcal{H} = \mathrm{InvGamma}(a, b)$, the exact marginal distribution of $(f_\infty^\infty(\mathbf{x}_{K+i}))_{i\in[L]}$ has the following form by Lemma B.2:

$$\mathrm{MVT}\left( 2\alpha, \mu', \frac{b}{a} \Theta' \right).$$

$\square$

**Lemma B.4.** *Let $(X_n)_{n\in\mathbb{N}}$ be a sequence of $p$-dimensional random vectors, $X$ be a $p$-dimensional random variable, and $Y$ be a random variable. If $X_n | Y$ converges in distribution to $X | Y$ almost surely (with respect to the randomness of $Y$) as $n$ tends to $\infty$, then $X_n$ converges in distribution to $X$.*

*Proof.* Let $h : \mathbb{R}^p \to \mathbb{R}$ be a bounded continuous function. We have to show that

$$\lim_{n \to \infty} \mathbb{E}[h(X_n)] = \mathbb{E}[h(X)].$$

The following calculation shows this equality:

$$\lim_{n \to \infty} \mathbb{E}[h(X_n)] = \lim_{n \to \infty} \mathbb{E}[\,\mathbb{E}[h(X_n) \mid Y]] = \mathbb{E}\left[\lim_{n \to \infty} \mathbb{E}[h(X_n) \mid Y]\right] = \mathbb{E}[\,\mathbb{E}[h(X) \mid Y]]$$
$$= \mathbb{E}[h(X)].$$

The first and last equalities follow from the standard tower law for conditional expectation, the second equality uses the dominated convergence theorem and the boundness of $h$, and the third equality uses the fact that $X_n \mid Y$ converges in distribution to $X|Y$ almost surely. $\square$

## C  STOCHASTIC VARIATIONAL GAUSSIAN PROCESS

Let $(X, \mathbf{y})$ be training points and $(X_*, y_*)$ be test points. And let Z be inducing points. We want to make $q(f, f_Z)$ which well approximates $p(f, f_Z|y)$ by maximizing ELBO. Then we will construct $q(f, f_Z) = p(f|f_Z)q(f_Z)$ where $p(f|f_Z) = \mathcal{N}(f|K_{XZ}K_{ZZ}^{-1}f_Z, K_{XX} - K_{XZ}K_{ZZ}^{-1}K_{ZX})$ and $q(f_Z) = \mathcal{N}(f_Z|\mu, \Sigma)$. Then

$$\log p(y) \geq \mathbb{E}_{f,f_Z \sim q(f,f_Z)}[\log p(y|f, f_Z)] - \mathrm{KL}(q(f_Z)||p(f_Z)).$$

Here we can calculate likelihood term as

$$\int \log p(y|f)\mathcal{N}(f; A\mu, A\Sigma A^\top + D)\mathrm{d}f$$

where $A = K_{XZ}K_{ZZ}^{-1}, B = K_{XX} - K_{XZ}K_{ZZ}^{-1}K_{ZX}$.
Also we can calculate KL term as

$$\frac{1}{2}\left[\log(\frac{\det K_{ZZ}}{\det \Sigma}) - n_Z + \mathrm{Tr}\{K_{ZZ}^{-1}\Sigma\} + \mu^\top K_{ZZ}^{-1}\mu\right].$$

We find optimal $\mu, \Sigma$ and inducing points $Z$ by using stochastic gradient descent. And with these optimal values, we can calculate predictive distribution $p(f_*|y)$.

$$p(f_*|y) = \int \int p(f_*, f, f_Z|y)\mathrm{d}f\mathrm{d}f_Z$$
$$= \int \int p(f_*|f, f_Z)q(f, f_Z)\mathrm{d}f\mathrm{d}f_Z$$
$$= \int \int p(f_*|f, f_Z)p(f|f_Z)q(f_Z)\mathrm{d}f\mathrm{d}f_Z$$
$$= \int p(f_*|f_Z)q(f_Z)\mathrm{d}f_Z.$$

This equation can be write as $p(f_*|y) = \mathcal{N}(f_*; K_{*Z}K_{ZZ}^{-1}\mu, K_{*Z}K_{ZZ}^{-1}\Sigma(K_{*Z}K_{ZZ}^{-1})^\top + K_{**} - K_{*Z}K_{ZZ}^{-1}K_{*Z}^\top)$.
Now if we use reparametrization trick at $f_Z$, we can write $f_Z = Lu$, where $L$ is the lower triangular matrix from the cholesky decomposition of the matrix $K_{ZZ}$. Then

$$p(f_Z) = \mathcal{N}(f_Z; 0, LL^\top) = \mathcal{N}(f_Z; 0, K_{ZZ})$$
$$q(f_Z) = \mathcal{N}(f_Z; L\mu_u, L\Sigma_u L^\top).$$

In this case ELBO changes into

$$\int \log(p(y|f))\mathcal{N}(f; AL\mu_u, AL\Sigma_u L^\top A^\top + B)\mathrm{d}f - \mathrm{KL}(\mathcal{N}(u; \mu_u, \Sigma_u)||\mathcal{N}(u; 0, I)).$$

With softmax likelihood, we can use MC approximation which is

$$\mathrm{ELBO} \simeq \frac{1}{T}\frac{N}{B}\sum_{t=1}^{T}\sum_{i=1}^{B}\log p(y_i|f_i^{(t)}) - \mathrm{KL}(\mathcal{N}(u; \mu_u, \Sigma_u)||\mathcal{N}(u; 0, I))$$

where $B$ is minibatch and $T$ is sample number of $f$ which sampled from

$$\mathcal{N}(f; A_B L \mu_u, A_B L \Sigma_u L^\top A_B^\top + B_B).$$

Here $A_B = K_{X_B Z} K_{ZZ}^{-1}$ and $B_B = K_{X_B X_B} - K_{X_B Z} K_{ZZ}^{-1} K_{Z X_B}$. Finally we can calculate predictive distribution with this reparametrization trick.

$$
\begin{aligned}
p(f_*|y) &= \int p(f_*|f_Z) q(f_Z) \mathrm{d}f_Z \\
&= \int \mathcal{N}(f_*; K_{*Z} K_{ZZ}^{-1} f_Z, K_{**} - K_{*Z} K_{*Z}^{-1} K_{Z*}) \mathcal{N}(f_Z; L\mu_u, L\Sigma_u L^\top) \mathrm{d}f_Z \\
&= \mathcal{N}(f_*; K_{*Z} K_{ZZ}^{-1} L\mu_u, K_{*Z} K_{ZZ}^{-1} L\Sigma_u L^\top (K_{*Z} K_{ZZ}^{-1})^\top + K_{**} - K_{*Z} K_{ZZ}^{-1} K_{*Z}^\top)
\end{aligned}
$$

Thus the final predictive distribution $p(y_*|x_*)$ is

$$
\begin{aligned}
\log p(y_*|x_*) &= \log \prod_j^M p(y_*^j | x_*^j) \\
&= \sum_j^M \log p(y_*^j | x_*^j) \\
&= \sum_j^M \log \int p(y_*^j | f_*) p(f_*|y) \mathrm{d}f_* \\
&\simeq \sum_j^M \log \frac{1}{N} \sum_i^N p(y_*^j | f_*^i)
\end{aligned}
$$

where $f_*^i$s are sampled from $\mathcal{N}(f_*; K_{*Z} K_{ZZ}^{-1} L\mu_u, K_{*Z} K_{ZZ}^{-1} L\Sigma_u L^\top (K_{*Z} K_{ZZ}^{-1})^\top + K_{**} - K_{*Z} K_{ZZ}^{-1} K_{*Z}^\top)$.

## D  STOCHASTIC VARIATIONAL STUDENT $t$ PROCESS

We want to make $q(f, f_Z, \sigma^2)$ which well approximates $p(f, f_Z, \sigma^2|y)$ by maximizing ELBO. First $p(f, f_Z, \sigma^2) = p(f|f_Z, \sigma^2) p(f_Z|\sigma^2) p(\sigma^2)$ where $p(f|f_Z, \sigma^2) = \mathcal{N}(f|K_{XZ} K_{ZZ}^{-1} f_Z, \sigma^2(K_{XX} - K_{XZ} K_{ZZ}^{-1} K_{ZX}))$, $p(f_Z|\sigma^2) = \mathcal{N}(f_Z|0, \sigma_2 K_{ZZ})$ and $p(\sigma^2) = \Gamma^{-1}(\sigma^2|\alpha, \beta)$. And $q(f, f_Z, \sigma^2) = p(f|f_Z, \sigma^2) q(f_Z|\sigma^2) q(\sigma^2)$ where $q(f_Z|\sigma^2) = \mathcal{N}(f_Z|\mu, \sigma^2 \Sigma)$ and $q(\sigma^2) = \Gamma^{-1}(\sigma^2|a, b)$. Then ELBO can be computed as

$$\log p(y) \geq \mathbb{E}_{f, f_Z, \sigma^2 \sim q(f, f_Z, \sigma^2)}[\log p(y|f, f_Z, \sigma^2)] - \mathrm{KL}(q(f, f_Z, \sigma^2)||p(f, f_Z, \sigma^2)).$$

Here KL divergence can be reformulated as

$$
\begin{aligned}
\mathrm{KL}(q(f, f_Z, \sigma^2)||p(f, f_Z, \sigma^2)) &= \int \int q(f_Z, \sigma^2) \log \frac{q(f_Z, \sigma^2)}{p(f_Z, \sigma^2)} \mathrm{d}f_Z \mathrm{d}\sigma^2 \\
&= \mathrm{KL}(q(f_Z, \sigma^2)||p(f_Z, \sigma^2)).
\end{aligned}
$$

Now let's compute likelihood part first.

$$\mathbb{E}_{f, f_Z, \sigma^2 \sim q(f, f_Z, \sigma^2)}[\log p(y|f, f_Z, \sigma^2)] = \int q(f) \log p(y|f) \mathrm{d}f$$

where $q(f) = \int \int p(f|f_Z, \sigma^2) q(f_Z|\sigma^2) q(\sigma^2) \mathrm{d}f_Z \mathrm{d}\sigma^2$. And this is

$$q(f) = \mathrm{MVT}(f|2a, K_{XZ} K_{ZZ}^{-1} \mu, \frac{b}{a}(K_{XZ} K_{ZZ}^{-1} \Sigma K_{ZZ}^{-1} K_{ZX} + K_{XX} - K_{XZ} K_{ZZ}^{-1} K_{ZX})).$$

By slightly modifying the KL divergence between normal-gamma distributions in Soch & Allefeld (2016), we get

$$\mathrm{KL}(q(f_Z, \sigma^2)||p(f_Z, \sigma^2)) = \int \int q(f_Z, \sigma^2) \log \frac{q(f_Z, \sigma^2)}{p(f_Z, \sigma^2)} \mathrm{d}f_Z \mathrm{d}\sigma^2$$

$$= \frac{1}{2}\frac{a}{b}\mu^\top K_{ZZ}^{-1}\mu + \frac{1}{2}\mathrm{Tr}(K_{ZZ}^{-1}\Sigma) + \frac{1}{2}\log\frac{|K_{ZZ}|}{|\Sigma|} - \frac{n_Z}{2} + \alpha\log\frac{b}{\beta}$$

$$- \log\frac{\Gamma(a)}{\Gamma(\alpha)} + (a - \alpha)\psi(a) + (\beta - b)\frac{a}{b}$$

where $\psi(\cdot)$ is digamma function. Now if we use reparametrization trick at $f_Z$, we can write $f_Z = Lu$, where $L$ is the lower triangular matrix from the cholesky decomposition of the matrix $K_{ZZ}$. Then

$$p(f_Z|\sigma^2) = \mathcal{N}(f_Z; 0, \sigma^2 LL^\top) = \mathcal{N}(f_Z; 0, \sigma^2 K_{ZZ})$$
$$q(f_Z|\sigma^2) = \mathcal{N}(f_Z; L\mu_u, \sigma^2 L\Sigma_u L^\top).$$

In this case ELBO changes into

$$\int \log(p(y|f))\mathrm{MVT}(f; 2a, AL\mu_u, \frac{b}{a}(AL\Sigma_u L^\top A^\top + B))\mathrm{d}f$$
$$- \mathrm{KL}(\mathcal{N}(u; \mu_u, \sigma^2\Sigma_u)\Gamma^{-1}(\sigma^2; a, b)||\mathcal{N}(u; 0, \sigma^2 I)\Gamma^{-1}(\sigma^2, \alpha, \beta)).$$

With softmax likelihood, we can use MC approximation which is

$$\mathrm{ELBO} \simeq \frac{1}{T}\frac{N}{B}\sum_{t=1}^{T}\sum_{i=1}^{B}\log p(y_i|f_i^{(t)})$$
$$- \mathrm{KL}(\mathcal{N}(u; \mu_u, \sigma^2\Sigma_u)\Gamma^{-1}(\sigma^2; a, b)||\mathcal{N}(u; 0, \sigma^2 I)\Gamma^{-1}(\sigma^2, \alpha, \beta))$$

where $B$ is minibatch and $T$ is sample number of $f$ which sampled from

$$\mathrm{MVT}(f; 2a, A_B L\mu_u, \frac{b}{a}(A_B L\Sigma_u L^\top A_B^\top + B_B)).$$

Here $A_B = K_{X_B Z}K_{ZZ}^{-1}$ and $B_B = K_{X_B X_B} - K_{X_B Z}K_{ZZ}^{-1}K_{ZX_B}$. Finally we can calculate predictive distribution with this reparametrization trick.

$$p(f_*|y) = \int \int p(f_*|f_Z, \sigma^2)q(f_Z|\sigma^2)q(\sigma^2)\mathrm{d}f_Z\mathrm{d}\sigma^2$$

$$= \int \int \mathcal{N}(f_*; K_{*Z}K_{ZZ}^{-1}f_Z, \sigma^2(K_{**} - K_{*Z}K_{*Z}^{-1}K_{Z*}))\mathcal{N}(f_Z; L\mu_u, \sigma^2(L\Sigma_u L^\top))\Gamma^{-1}(a, b)\mathrm{d}f_Z\mathrm{d}\sigma^2$$

$$= \mathrm{MVT}(f_*; 2a, K_{*Z}K_{ZZ}^{-1}L\mu_u, \frac{b}{a}(K_{*Z}K_{ZZ}^{-1}L\Sigma_u L^\top(K_{*Z}K_{ZZ}^{-1})^\top + K_{**} - K_{*Z}K_{ZZ}^{-1}K_{*Z}^\top)).$$

Thus the final predictive distribution $p(y_*|x_*)$ is

$$\log p(y_*|x_*) = \log \prod_j^M p(y_*^j|x_*^j)$$

$$= \sum_j^M \log p(y_*^j|x_*^j)$$

$$= \sum_j^M \log \int p(y_*^j|f_*)p(f_*|y)\mathrm{d}f_*$$

$$\simeq \sum_j^M \log \frac{1}{N}\sum_i^N p(y_*^j|f_*^i)$$

where $f_*^i$s are sampled from $\mathrm{MVT}(f_*; 2a, K_{*Z}K_{ZZ}^{-1}L\mu_u, \frac{b}{a}(K_{*Z}K_{ZZ}^{-1}L\Sigma_u L^\top(K_{*Z}K_{ZZ}^{-1})^\top + K_{**} - K_{*Z}K_{ZZ}^{-1}K_{*Z}^\top)).$

# E INFERENCE ALGORITHMS

## E.1 INFERENCE FOR REGRESSION

In order to calculate the predictive posterior distribution for an intractable marginal distribution, we can use self-normalized importance sampling. Consider a scale mixture of NNGPs with a prior $\mathcal{H}$ on the variance $\sigma_v^2$. Assume that we would like to estimate the expectation of $h(y)$ for some function $h : \mathbb{R} \to \mathbb{R}$ where the random variable $y$ is drawn from the predictive posterior of the mixture at some input $\mathbf{x} \in \mathbb{R}^I$ under the condition on $\mathcal{D}_{\text{tr}}$. Then, the expectation of $h(y)$ is

$$
\begin{aligned}
\mathbb{E}[h(y)] &= \int h(y) p(y|\mathcal{D}_{\text{tr}}) \mathrm{d}y \\
&= \int \int h(y) p(y|\mathcal{D}_{\text{tr}}, \sigma_v^2) p(\sigma_v^2|\mathcal{D}_{\text{tr}}) \mathrm{d}\sigma_v^2 \mathrm{d}y \\
&= \int \int h(y) p(y|\mathcal{D}_{\text{tr}}, \sigma_v^2) \frac{p(\sigma_v^2, Y_{\text{tr}}|X_{\text{tr}})}{p(Y_{\text{tr}}|X_{\text{tr}})} \mathrm{d}\sigma_v^2 \mathrm{d}y \\
&= \frac{1}{Z} \int \int h(y) p(y|\mathcal{D}_{\text{tr}}, \sigma_v^2) \frac{p(\sigma_v^2, Y_{\text{tr}}|X_{\text{tr}})}{p(\sigma_v^2)} p(\sigma_v^2) \mathrm{d}\sigma_v^2 \mathrm{d}y
\end{aligned}
$$

where $Z = \int p(\sigma_v^2, Y_{\text{tr}}|X_{\text{tr}}) \mathrm{d}\sigma_v^2$. We can approximate $Z$ as follows:

$$
\begin{aligned}
Z &= \int \frac{p(\sigma_v^2, Y_{\text{tr}}|X_{\text{tr}})}{p(\sigma_v^2)} p(\sigma_v^2) \mathrm{d}\sigma_v^2 \\
&\simeq \frac{1}{N} \sum_{i=1}^{N} \frac{p(\beta_i, Y_{\text{tr}}|X_{\text{tr}})}{p(\beta_i)} \\
&= \frac{1}{N} \sum_{i=1}^{N} p(Y_{\text{tr}}|X_{\text{tr}}, \beta_i)
\end{aligned}
$$

where $\beta_i$s are sampled independently from $\mathcal{H}$. Using this approximation, we can also approximate expectation of $h(y)$ as follows:

$$
\begin{aligned}
\mathbb{E}[h(y)] &= \frac{1}{Z} \int \int h(y) p(y|\mathcal{D}_{\text{tr}}, \sigma_v^2) \frac{p(\sigma_v^2, Y_{\text{tr}}|X_{\text{tr}})}{p(\sigma_v^2)} p(\sigma_v^2) \mathrm{d}\sigma_v^2 \mathrm{d}y \\
&\simeq \sum_{i=1}^{N} \frac{w_i}{\sum_{j=1}^{N} w_j} h(y_i)
\end{aligned}
$$

where the $y_i$'s are sampled from the posterior of the Gaussian distribution and the $w_i$'s are the corresponding importance weights for all $i \in [N]$:

$$
\beta_i \sim \mathcal{H}, \quad w_i = \mathcal{N}(Y_{\text{tr}}; 0, \beta_i \overline{\mathcal{K}}_{\text{tr,tr}}), \quad y_i \sim \mathcal{N}(\overline{\mathcal{K}}_{\mathbf{x},\text{tr}} \overline{\mathcal{K}}_{\text{tr,tr}}^{-1} Y_{\text{tr}}, \beta_i (\overline{\mathcal{K}}_{\mathbf{x},\mathbf{x}} - \overline{\mathcal{K}}_{\mathbf{x},\text{tr}} \overline{\mathcal{K}}_{\text{tr,tr}}^{-1} \overline{\mathcal{K}}_{\text{tr},\mathbf{x}})).
$$

The $\overline{\mathcal{K}}$ is the covariance matrix computed with test input as in Theorem 3.1. To speed up calculation, we removed duplicated calculations in our importance weights and also during the sampling of the $y_i$'s. First, for importance weights, we observe that the log likelihood of $\beta_i$ has the form:

$$
\log \mathcal{N}(Y_{\text{tr}}; 0, \beta_i \overline{\mathcal{K}}_{\text{tr,tr}}) = -\frac{K}{2} \log(2\pi) - \frac{1}{2} \log \det(\overline{\mathcal{K}}_{\text{tr,tr}}) - \frac{K}{2} \log(\beta_i) - \frac{1}{2\beta_i} Y_{\text{tr}}^{\top} \overline{\mathcal{K}}_{\text{tr,tr}}^{-1} Y_{\text{tr}},
$$

and the three terms $\frac{K}{2} \log(2\pi)$, $\frac{1}{2} \log \det(\mathcal{K}_{\text{tr,tr}})$, and $Y_{\text{tr}}^{\top} \overline{\mathcal{K}}_{\text{tr,tr}}^{-1} Y_{\text{tr}}$ here are independent with $\beta_i$. Thus, using this observation, we compute these three terms beforehand only once, and calculate the log likelihood of each sample $\beta_i$ in $O(1)$ time. Next, for sampling the $y_i$'s, we first draw $N$ samples $\bar{y}_i$s from $\mathcal{N}(0, \overline{\mathcal{K}}_{\mathbf{x},\mathbf{x}} - \overline{\mathcal{K}}_{\mathbf{x},\text{tr}} \overline{\mathcal{K}}_{\text{tr,tr}}^{-1} \overline{\mathcal{K}}_{\text{tr},\mathbf{x}})$, then multiply each of these samples with $\sqrt{\beta_i}$, and finally add $\overline{\mathcal{K}}_{\mathbf{x},\text{tr}} \overline{\mathcal{K}}_{\text{tr,tr}}^{-1} Y_{\text{tr}}$ to each of the results. Since $\overline{\mathcal{K}}_{\mathbf{x},\text{tr}} \overline{\mathcal{K}}_{\text{tr,tr}}^{-1} Y_{\text{tr}} + \sqrt{\beta_i} \bar{y}_i \sim \mathcal{N}(\overline{\mathcal{K}}_{\mathbf{x},\text{tr}} \overline{\mathcal{K}}_{\text{tr,tr}}^{-1} Y_{\text{tr}}, \beta_i (\overline{\mathcal{K}}_{\mathbf{x},\mathbf{x}} - \overline{\mathcal{K}}_{\mathbf{x},\text{tr}} \overline{\mathcal{K}}_{\text{tr,tr}}^{-1} \overline{\mathcal{K}}_{\text{tr},\mathbf{x}}))$ for all $i \in [N]$, we can use these final outcomes of these computations as the $y_i$'s.

**Table 4:** Experimental results of Classification with Gaussian Likelihood

| Dataset | Accuracy | NNGP | Inverse Gamma | Burr Type XII |
|---|---|---|---|---|
| MNIST | $96.8 \pm 0.2$ | $9.29 \pm 0.002$ | $\mathbf{4.89} \pm 0.001$ | $4.96 \pm 0.001$ |
| + Shot Noise | $94.9 \pm 0.4$ | $9.32 \pm 0.001$ | $\mathbf{4.88} \pm 0.001$ | $4.96 \pm 0.001$ |
| + Impulse Noise | $84.4 \pm 2.3$ | $9.63 \pm 0.000$ | $\mathbf{5.17} \pm 0.001$ | $5.25 \pm 0.001$ |
| + Spatter | $95.4 \pm 0.4$ | $9.27 \pm 0.001$ | $\mathbf{4.44} \pm 0.002$ | $4.49 \pm 0.007$ |
| + Glass Blur | $90.5 \pm 0.7$ | $9.12 \pm 0.001$ | $4.20 \pm 0.005$ | $\mathbf{4.00} \pm 0.017$ |
| + Zigzag | $84.9 \pm 1.5$ | $9.51 \pm 0.001$ | $4.62 \pm 0.006$ | $\mathbf{4.49} \pm 0.021$ |
| EMNIST | $70.4 \pm 0.9$ | $9.40 \pm 0.001$ | $\mathbf{4.15} \pm 0.004$ | $4.45 \pm 0.013$ |
| Fashion MNIST | $61.3 \pm 5.1$ | $9.34 \pm 0.002$ | $\mathbf{4.96} \pm 0.002$ | $4.97 \pm 0.002$ |
| KMNIST | $81.1 \pm 1.3$ | $9.48 \pm 0.001$ | $4.60 \pm 0.002$ | $\mathbf{4.34} \pm 0.013$ |
| SVHN | $42.5 \pm 1.5$ | $6.01 \pm 0.062$ | $4.16 \pm 0.013$ | $\mathbf{4.15} \pm 0.009$ |

### E.2 TIME COMPLEXITY ANALYSIS

For time complexity, as we mentioned in Appendix E, our posterior-predictive algorithm based on importance sampling does not induce significant overhead thanks to the reuse of the shared terms for calculation. More specifically, our algorithm with K sample variances spends $O(K + N^3)$ time, instead of $O(KN^3)$, for computing a posterior predictive, where N is the number of training points. Compare this with the usual time complexity $O(N^3)$ of the standard algorithm for Gaussian processes. When it comes to SVGP and SVTP, one update step of both SVGP and SVTP takes $O(BM^2 + M^3)$ time, where B is the number of the batch size of the input dataset and M is the number of the inducing points.

## F CLASSIFICATION WITH GAUSSIAN LIKELIHOOD

We compare the NNGPs and the scale mixture of NNGPs with Inverse Gamma prior and Burr Type XII prior for the classification tasks. Following the convention (Lee et al., 2018; Novak et al., 2018; Garriga-Alonso et al., 2019), we treat the classification problem as a regression problem where the regression targets are one-hot vectors of class labels and computed the posteriors induced from squared-loss. We searched hyperparameters refer to Adlam et al. (2020) and we choose both $c$ and $k$ from $[0.5, 1., 2., 3., 4.]$. We do not particularly expect the scale-mixtures of NNGPs to outperform NNGP in terms of predictive accuracy, but we expect them to excel in terms of calibration due to its flexibility in describing the variances of class probabilities. To better highlight this aspect, for MNIST data, we trained the models on clean training data but tested on corrupted data to see how the models would react under such distributional shifts. The results are summarized in Table 4. Due to the limitation of the resources for computing full data kernel matrix, we use 5000 samples as train set, and 1000 samples as test set. For all datasets, the scale-mixture of NNGPs outperform NNGP in terms of NLL.

## G EXPERIMENTAL DETAILS

In Fig. 1, we used the inverse gamma prior with hyperparameter setting $\alpha = 2$ and $\beta = 2$ for the experiments validating convergence of initial distribution (Theorem 3.1) and last layer training (Theorem 3.2). For the full layer training (Theorem 3.3), we used $\alpha = 1$ and $\beta = 1$.

For the regression experiments, we divide each dataset into train/validation/test sets with the ratio $0.8/0.1/0.1$. We performed gradient descent in order to update parameters of our models except number of layers which is discrete value. We referred Adlam et al. (2020) for initializing parameters of NNGPs. We choose the best hyperparameters based on validation NLL values and measured NLL values on the test set with permuted train sets.

For the classification experiments, we divide each dataset into train/test sets as provided by TensorFlow Datasets[2], and further divide train sets into train/validation sets with the ratio $0.9/0.1$. We choose the best hyperparameters based on the validation NLL values, and measure NLL and accuracy values on the test set with different initialization seeds. To measure the uncertainty calibration

---

[2]https://www.tensorflow.org/datasets

**Table 5:** RMSE values on UCI dataset. (m, d) denotes number of data points and features, respectively. We take results from Adlam et al. (2020) except our model.

| Dataset | $(m, d)$ | PBP-MV | Dropout | Ensembles | RBF | NNGP | Ours |
|---|---|---|---|---|---|---|---|
| Boston Housing | (506, 13) | $3.11 \pm 0.15$ | $\mathbf{2.90} \pm 0.18$ | $3.28 \pm 1.00$ | $3.24 \pm 0.21$ | $3.07 \pm 0.24$ | $3.30 \pm 0.03$ |
| Concrete Strength | (1030, 8) | $5.08 \pm 0.14$ | $\mathbf{4.82} \pm 0.16$ | $6.03 \pm 0.58$ | $5.63 \pm 0.24$ | $5.25 \pm 0.20$ | $5.08 \pm 0.14$ |
| Energy Efficiency | (768, 8) | $0.45 \pm 0.01$ | $0.54 \pm 0.06$ | $2.09 \pm 0.29$ | $0.50 \pm 0.01$ | $0.57 \pm 0.02$ | $\mathbf{0.44} \pm 0.03$ |
| Kin8nm | (8192, 8) | $\mathbf{0.07} \pm 0.00$ | $0.08 \pm 0.00$ | $0.09 \pm 0.00$ | $\mathbf{0.07} \pm 0.00$ | $\mathbf{0.07} \pm 0.00$ | $\mathbf{0.07} \pm 0.00$ |
| Naval Propulsion | (11934, 16) | $\mathbf{0.00} \pm 0.00$ | $\mathbf{0.00} \pm 0.00$ | $\mathbf{0.00} \pm 0.00$ | $\mathbf{0.00} \pm 0.00$ | $\mathbf{0.00} \pm 0.00$ | $\mathbf{0.00} \pm 0.00$ |
| Power Plant | (9568, 4) | $3.91 \pm 0.04$ | $4.01 \pm 0.04$ | $4.11 \pm 0.17$ | $3.82 \pm 0.04$ | $3.61 \pm 0.04$ | $\mathbf{3.53} \pm 0.04$ |
| Wine Quality Red | (1588, 11) | $0.64 \pm 0.01$ | $0.62 \pm 0.01$ | $0.64 \pm 0.04$ | $0.64 \pm 0.01$ | $\mathbf{0.57} \pm 0.01$ | $0.59 \pm 0.01$ |
| Yacht Hydrodynamics | (308, 6) | $0.81 \pm 0.06$ | $0.67 \pm 0.05$ | $1.58 \pm 0.48$ | $0.60 \pm 0.07$ | $0.41 \pm 0.04$ | $\mathbf{0.35} \pm 0.04$ |

**Table 6:** Additional regression results for Burr Type XII prior distribution.

| Dataset | $(m, d)$ | NNGP | Inverse Gamma prior | Burr Type XII prior |
|---|---|---|---|---|
| Boston Housing | (506, 13) | $\mathbf{2.65} \pm 0.13$ | $2.72 \pm 0.05$ | $2.77 \pm 0.02$ |
| Concrete Strength | (1030, 8) | $3.19 \pm 0.05$ | $\mathbf{3.13} \pm 0.04$ | $3.29 \pm 0.09$ |
| Energy Efficiency | (768, 8) | $1.01 \pm 0.04$ | $\mathbf{0.67} \pm 0.04$ | $0.70 \pm 0.04$ |
| Kin8nm | (8192, 8) | $-1.15 \pm 0.01$ | $-1.18 \pm 0.01$ | $\mathbf{-1.23} \pm 0.01$ |
| Naval Propulsion | (11934, 16) | $\mathbf{-10.01} \pm 0.01$ | $-8.04 \pm 0.04$ | $-4.38 \pm 0.05$ |
| Power Plant | (9568, 4) | $2.77 \pm 0.02$ | $\mathbf{2.66} \pm 0.01$ | $2.78 \pm 0.01$ |
| Wine Quality Red | (1588, 11) | $\mathbf{-0.98} \pm 0.06$ | $-0.77 \pm 0.07$ | $-0.16 \pm 0.05$ |
| Yacht Hydrodynamics | (308, 6) | $1.07 \pm 0.27$ | $\mathbf{0.17} \pm 0.25$ | $0.60 \pm 0.15$ |

of the model, we used the corrupted variants of the dataset and made three new dataset from the base datasets. For the corrupted variants, we trained the model on the original version of the datasets, and tested on the provided corrupted verions. For the out-of-distribution variants, we removed 3 classses on the train set. For imbalance variants, we limited the samples per class with exponential ratio on the train set. For noisy label variants, we selected 50% of labels and assigned uniformly selected new labels from the train set. Except the corrupted variants, we tested the model on the original test set.

For the classfication by categorical likelihood experiments, we use the CNN model with two layers to compute the kernel. Due to the limitations of computing resources, we can only use 400 inducing points for the variational inference which leads slight degradation on the performance. For the classification by finite model ensemble experiments, each CNN layer has 128 channels.

# H    ADDITIONAL EXPERIMENTS

The experiment code is available at GitHub[3]. We used a server with Intel Xeon Silver 4214R CPU and 128GB of RAM to evaluate the classification with Gaussian likelihood experiment, and used NVIDIA GeForce RTX 2080Ti GPU to conduct other experiments.

**Impact of the prior hyperparameters**    With same experimental setting with Section 4.1, in order to inspect the impact of the prior hyperparameters, we sampled 1000 models for initial, last layer training and full layer training. In Fig. 2, here we can empirically see that consistently with the theory, if $\alpha$ is smaller, the output distribution is more heavy-tailed.

**Full-layer training correspondence for ResNet**    Even though we only proved the general training result (Theorem 3.3) only for the fully-connected neural networks, we experimentally found that ResNet also shows a behavior predicted by our theorem. The empirical validation with for ResNet is shown in Fig. 3. We set $\alpha = 4$ and $\beta = 4$ for the inverse gamma prior in this experiment.

**Additional results of regression with Gaussian likelihood**    In addition to Table 1, we also measured the root mean square error values on the test set. Table 5 summarizes the results. The results other than ours are borrowed from Adlam et al. (2020) and we use same settings as described in Section 4.2.

---

[3]https://github.com/Anonymous0109/Scale-Mixtures-of-NNGP

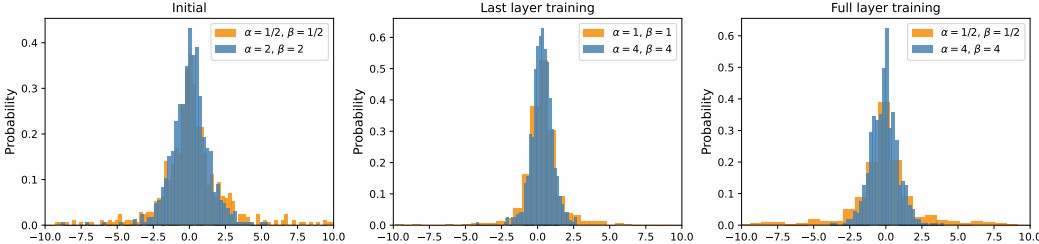

**Figure 2:** Impact of the prior hyperparameters for fully connected neural network initial, last layer training and full layer training.

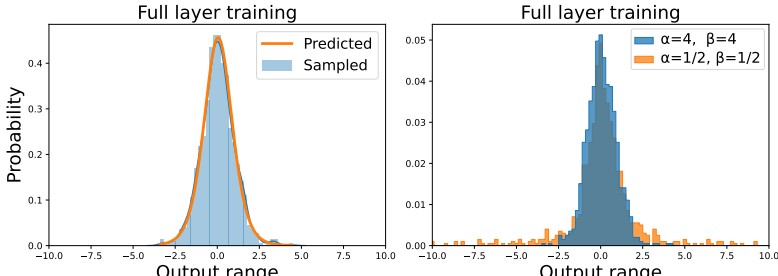

**Figure 3:** (left) Correspondence between wide finite ResNet model vs theoretical limit. (right) impact of the prior hyperparameters.

**Additional regression experiment** As an additional regression experiments, we test the models which use Burr Type XII distribution as its last layer variance's prior. We use Appendix E in order to calculate the predictive posterior distribution. Our results are in Table 6.

# I ADDITIONAL INFORMATION

We refer to the source of each dataset through footnotes with URL links. We don't use any data which obtained from people and don't use any data which contains personally identifiable information or offensive content.

