# OpenReview forum: "Scale Mixtures of Neural Network Gaussian Processes"
_ICLR.cc/2022/Conference — ICLR 2022 Poster_

### Official Review · Reviewer_C7Q4 · 2021-10-26

**Correctness:** 4
**Technical Novelty And Significance:** 1
**Empirical Novelty And Significance:** 1
**Recommendation:** 5
**Confidence:** 4

**Main Review:**

Font in tables is too small to be readable. Same for Fig. 1.

The paper in general clear and well written. The experiments are also extensive. The theorems seem correct although I did not check them thoroughly.

My main point of criticism of this paper is that there is little novelty in it. Although it provides interesting theoretical results, the main application, i.e., the use of Student's T process is already known in the machine learning community. So there are little new practical applications of this work.

Summing up, I believe that although this paper is interesting it is not expected to have a big impact in the machine learning community.





**Summary Of The Paper:**

This paper proposes to extend the typical GP formulation as the limiting case of a NN with Gaussian weights when the number of hidden units tends to infinity. In particular, it considers a scale of mixture of Gaussians at the last layer for the prior. When this scale mixture of Gaussians is a Inv. Gamma distribution it is well known that the result is a Student-t process. The paper also gives some insights about the resulting process obtained by training the weights of the last layer and following a similar initialization. The resulting processes are heavy-tailed and more robust as shown in the experiments.


**Summary Of The Review:**

Nice theoretical results that lead to already known methods in the machine learning community which questions the novelty of the paper.

---

> ### Author Response · Authors · 2021-11-13
> **Response to Reviewer C7Q4**
>
> We thank you for the constructive comments.
>
> Our work is not limited to the Student $t$ process. We show the relationship between infinite-width neural networks and heavy-tail processes. As we have mentioned in the paper, our work extends the existing infinite-width limits of neural networks to much more flexible classes of stochastic processes, with a simple easy-to-use modification of placing a prior distribution only on the scale of the readout layer. Moreover, despite the increased flexibility, our work does not lose most of the desirable properties of NNGP; easy-to-use posterior inference algorithms and clean analysis of gradient-descent training. This benefit can also be seen when we compare our work with that of Favaro et al. (2020) and Bracale et al. (2021), where alternative prior distributions (Stable distributions) with carefully chosen parameterization were put on the entire layers. In (Favaro et al., 2020), the infinite limit of such parameterization does not admit a closed-form posterior nor an easy-to-implement posterior inference algorithm.And our framework supports a large class of practically important stable processes as special cases. Note also that we proposed a stochastic variational Student $t$ process based on the scale mixture of Gaussian processes with inverse gamma prior.
>
> (Favaro et al., 2020) Favaro, S., Fortini, S., and Stefano, P. Stable behaviour of infinitely wide deep neural networks. AISTATS, 2020.
>
> (Bracale et al., 2021) Bracale, D., Favaro, S., Fortini, S., and Peluchetti, S.  Infinite-channel deep stable convolutional neural networks. arXiv preprint arXiv:2102.03739.

---

> > ### Comment · Reviewer_C7Q4 · 2021-11-24
> > **Response to Author's Rebuttal**
> >
> > I would like to thank the authors for the response provided to my review. I agree that the paper has new contributions from the theoretical point of view. However, from a practical point of view, I still believe that the novelty is low. The reason is that the proposed method is a Student-t process which is already known in the literature. While the authors propose a sparse Variational Student-t process for some of the problems, it is heavily based on the derivations for the traditional sparse GP. The other scale mixture used besides the inverse-gamma, i.e., the  Burr Type XII, does not seem to improve results over the inverse-gamma, which questions its practical utility. Therefore, I will prefer to keep my score as it is.

---

### Official Review · Reviewer_QZea · 2021-10-31

**Correctness:** 4
**Technical Novelty And Significance:** 2
**Empirical Novelty And Significance:** Not applicable
**Recommendation:** 6
**Confidence:** 2

**Main Review:**

Strengths:
- Interesting extension of NNGPs to consider limiting processes which can have heavy tails, backed by rigorous convergence results and empirical evaluations
- The paper is mostly well presented and easy to follow

Weaknesses:
- There are empirical findings showing that the scale mixture of NNGPs leads to robustness to the out-of-distribution data, but no theoretical results are provided to justify these (would be nice to at least give some intuition)

Comments/questions:
- It is not clear to me how and why the heavy tail nature of the NNGPs leads to the robustness (also, why not robustness to noisy labels, imbalance, etc.). What is the key intuition there?
- As for the empirical results, the scale mixture of NNGPs seems to give mixed performance (both accuracy and robustness to various perturbations) for different datasets. How should one decide which model to use for a given task to achieve desirable accuracy and robustness? Are there some trade-offs?

Minor comments:
- It may be a good idea to define the inverse Gamma distribution introduced on page 4
- There is a missing comma after the first equation in Theorem 3.2

**Summary Of The Paper:**

Building on the recent results on the correspondence between infinite-width neural networks and Gaussian processes (NNGPs), this paper proposes and studies a simple extension, looking at a scale mixture of such processes. The key idea is to introduce a scale prior on the parameters of the last layer, allowing construction of a richer classes of stochastic processes, particularly those with heavy tails. Some convergence results for these general processes are obtained by applying the tensor program and the Master Theorem, under various settings. Empirical results are also provided, showing the promise of the approach and robustness to out-of-distribution data.

**Summary Of The Review:**

Overall, this is a well written paper that studies an interesting extension of NNGPs to a scale mixture version, which allows realization of a richer class of stochastic processes such as those with heavy tails. Rigorous results are provided and they seem to be technically correct (although I have not checked all the details). These are the key strengths of the paper and the main factors behind my rating (an acceptance). However, no intuition or theory is provided to justify the empirical findings for the robustness, which seems to be an important practical utility of the proposed approach.

---

> ### Author Response · Authors · 2021-11-13
> **Response to Reviewer QZea**
>
> We thank you for your constructive comments.
>
> Intuition on why heavy-tailedness leads to robustness: as you pointed out, we don’t have a theoretical guarantee that a model with a heavy-tailed error distribution would be more robust than a model with a light-tailed one. But there is a heuristic informal explanation. A model with heavy-tailed error produces wider credible intervals, and thus is more conservative in its prediction. When encountered with an abnormal situation (out-of-distribution data, outliers, …), such a model would place more probability on being wrong and become less certain, leading to better calibrated prediction (which is well reflected in improved NLL values in our experiments). In fact, a similar observation and a rationale can be found in many existing works. For instance, (Yu et al., 2007) proposes to use $t$ processes for robust multi-task learning under the presence of outliers. (Jylänki et al., 2011) presents a robust Gaussian process regression method with student-t likelihood. (Shah et al., 2014) reports that Student $t$ processes are more robust than Gaussian processes to the change-points or model misspecification.
>
> (Yu et al., 2007) Yu, S., Tresp, V., and Yu, K.. Robust multi-task learning with $t$ processes. ICML 2007.
>
> (Jylänki et al., 2011) Jylänki, P., Vanhatalo, J., and Vehtari, A.. Robust Gaussian process regression with a Student-$t$ likelihood. JMLR, 2011.
>
> (Shah et al., 2014) Shah, A., Wilson, A. G., and Ghahramani, Z.. Student-$t$ processes as alternatives to Gaussian processes. AISTATS, 2014.
>
> Mixed performance of NNGP and scale mixture of NNGPs, when to use which one?: It may depend on the characteristics of the dataset. In fact, our model (especially with inverse gamma prior) contains NNGP as a special case of our model (both beta and alpha are infinite for inverse gamma case). The only additional overhead for this case is that we have extra hyperparameters (alpha and beta), but they can be learned via empirical Bayes. Hence, we would argue that one can safely use ours with inverse gamma as an alternative to NNGP for most problems in practice.
>
> Definition of inverse gamma: we have put the definition of the inverse gamma distribution in the appendix.
>
> Typo: thanks. Fixed.

---

> > ### Comment · Reviewer_QZea · 2021-11-25
> > **Acknowledgement of Author's Rebuttal**
> >
> > I would like to thank the author(s) for the response. Overall it is a good paper but I have the same concern as Reviewer C7Q4 does. Therefore, I am going to keep my score.

---

### Official Review · Reviewer_Z24X · 2021-11-02

**Correctness:** 3
**Technical Novelty And Significance:** 3
**Empirical Novelty And Significance:** Not applicable
**Recommendation:** 6
**Confidence:** 2

**Main Review:**

### Strengths

-	Tensor programs for expressing GPs using the NNGP formulation are a fairly recent development, and the work presented here is consequently a timely contribution to a topic that has accrued great interest within the community.
-	Although the core contribution feels fairly incremental in nature, both the execution and the corresponding analysis are non-trivial. Although I was not familiar enough with tensor programs to properly go through and understand all the derivations featured in the paper, the exposition appears to be quite thorough all throughout the paper, with sufficient detail also being provided in the supplementary material.
-	The experiments are broad and varied, covering a wide selection of possible practical challenges (noisy data/unreliable labels, etc), where the proposed technique might be particularly effective.

### Weaknesses

-	While well-written overall, I often found the paper to be quite difficult to follow properly. The paper contains a lot of set-up, whereas the paper’s own contributions are only presented from the very end of page 4 onwards. Even then, the majority of the paper’s contributions are all grouped together in a single ‘Results’ section that are only slightly disambiguated under three different theorems. Although the paper already contains a very extensive and thorough appendix, I do believe that the paper would benefit from some additional restructuring and refactoring.
-	It would also be nice to expand on some of the related work (in particular the competing approaches highlighted for CNNs) in order to better emphasise how the approach being proposed in this paper is ‘simpler’ while being just as effective.
-	I have a few questions on the Experiments section:
--	Is there a particular reason for not showing RMSE results for the regression datasets? While the benefits of using a heavier-tailed distribution may indeed be more pronounced in the MNLL metrics, it would still be interesting to see whether this comes at the expense of possibly inferior mean predictions.
--	For the results on the regression datasets shown in Table 1, the authors comment that the results for competing methods are copied from the work of Adlam et al (2020). However, given that there are no predefined train/test splits for these UCI datasets, how are you ensuring that the same splits are used as for this earlier work?
--	Would it be possible to compare against the other heavy-tailed stochastic processes described in the Related Work section? This should at least be possible for the image classification experiments right? There’s also a small error in the caption of Figure 1 which contains references to ‘top’ and ‘bottom’ figures.
-	There are a few typos remaining in the paper, but these should be fairly easy to with a proper read-through of the paper. A few of the titles in the references also require capitalization for words such as Gaussian, etc.


**Summary Of The Paper:**

Although the connection between Bayesian neural networks and Gaussian processes has been a topic of interest for several years, the work on tensor programs facilitating the formulation of the Neural Network-Gaussian Process (NNGP) model has been influential in further emphasising the correspondence between the two classes of models. In this work, the authors propose an extension to the NNGP model that allows for the formulation of more general stochastic processes that may follow alternative distributions such as Student-$t$. This is achieved by way of introducing a prior on the scale of the parameters in the last layer of the neural network. This formulation allows for greater flexibility without requiring the more involved changes proposed in related works such as Favran et al. (2020) and Bracale et al. (2021). Following the analysis carried out in the initial formulation of NNGPs, the authors also investigate the correspondence between GPs and infinitely-wide BNNs configured with the proposed set-up and trained using gradient descent. The experimental evaluation features a mixture of regression and classification tasks, with a particular emphasis on how the use of models with heavier-tailed distributions can be better suited to datasets with challenging properties such as corrupted data and label imbalance.

**Summary Of The Review:**

The contributions featured in this paper provide a valuable and non-trivial extension to the literature on NNGPs. Given that this is a fairly new area of research, such work is especially timely. Even so, I still believe that the paper’s writing could be heavily improved in order to make the exposition more clear. I also have some remaining concerns on the experimental evaluation that I would like to see addressed in either the rebuttal or a future revision of the paper before tending more heavily towards acceptance.

** Rebuttal Update **

Raised score to 6 following clarifications and additional experiment results provided by authors.

---

> ### Author Response · Authors · 2021-11-13
> **Response to Reviewer Z24X**
>
> Thank you for your constructive comments.
>
> Paper organization: although we couldn’t refactor the paper during the rebuttal period, we are planning to re-organize the paper and incorporate your suggestion; reduce the setup part in the main text by moving details to appendix, and highlight our results more in the main text.
>
> Expanding related work: we think the most related previous works (expanding the class of infinite-width neural networks) are the ones with stable processes (Favaro et al., 2020, Bracale et al., 2021). For those works, we have elaborated what we mean by “simple” in the revised version of the paper. See the highlighted sentences in the revised paper.
>
> Why no RMSE?: Thank you for pointing this out. We measured the RMSE and found that ours show good performance in terms of RMSE as well.  We put this result in the additional-experiment section in the appendix of the revised paper.
>
> | Dataset             |   PBP-MV        |   Dropout       |   Ensembles     |   RBF           |   NNGP          |   Ours          |
> | :------------------ | :-------------: | :-------------: | :-------------: | :-------------: | :-------------: | :-------------: |
> | Boston Housing      |   3.11 ± 0.15   | **2.90 ± 0.18** |   3.28 ± 1.00   |   3.24 ± 0.21   |   3.07 ± 0.24   |   3.30 ± 0.03   |
> | Concrete Strength   |   5.08 ± 0.14   | **4.82 ± 0.16** |   6.03 ± 0.58   |   5.63 ± 0.24   |   5.25 ± 0.20   |   5.08 ± 0.14   |
> | Energy Efficiency   |   0.45 ± 0.01   |   0.54 ± 0.06   |   2.09 ± 0.29   |   0.50 ± 0.01   |   0.57 ± 0.02   | **0.44 ± 0.03** |
> | Kin8nm              | **0.07 ± 0.00** |   0.08 ± 0.00   |   0.09 ± 0.00   | **0.07 ± 0.00** | **0.07 ± 0.00** | **0.07 ± 0.00** |
> | Naval Propulsion    | **0.00 ± 0.00** | **0.00 ± 0.00** | **0.00 ± 0.00** | **0.00 ± 0.00** | **0.00 ± 0.00** | **0.00 ± 0.00** |
> | Power Plant         |   3.91 ± 0.04   |   4.01 ± 0.04   |   4.11 ± 0.17   |   3.82 ± 0.04   |   3.61 ± 0.04   | **3.53 ± 0.04** |
> | Wine Quality Red    |   0.64 ± 0.01   |   0.62 ± 0.01   |   0.64 ± 0.04   |   0.64 ± 0.01   | **0.57 ± 0.01** |   0.59 ± 0.01   |
> | Yacht Hydrodynamics |   0.81 ± 0.06   |   0.67 ± 0.05   |   1.58 ± 0.48   |   0.60 ± 0.07   |   0.41 ± 0.04   | **0.35 ± 0.04** |
>
> Use of the same train/test split as in Adlam et al (2020): Adlam et al. (2020) also take results from Mukhoti et al. (2018) and Lakshminarayanan et al. (2017) except the RBF and NNGP results. As far as we know, the original paper did not make their random seeds public so it is impossible to perfectly reproduce their experimental setting. Thus, in our submission, we followed the previous work and borrowed the numbers from the original papers directly. During the rebuttal period, however, we did additional experiments evaluating the baselines using our own split that was used to evaluate our models; the performance gap was insignificant except for Naval propulsion dataset, for which our reproduced results actually performed worse than the copied results.
>
> | Dataset             | NNGP(Copied)  | NNGP(Reproduce) | Ours         |
> | :------------------ | :-----------: | :-------------: | :----------: |
> | Boston Housing      |   2.65 ± 0.13 |     2.65 ± 0.02 |  2.72 ± 0.05 |
> | Concrete Strength   |   3.19 ± 0.05 |     3.16 ± 0.01 |  3.13 ± 0.04 |
> | Energy Efficiency   |   1.01 ± 0.04 |     1.09 ± 0.06 |  0.67 ± 0.04 |
> | Kin8nm              |  -1.15 ± 0.01 |    -1.09 ± 0.00 | −1.18 ± 0.01 |
> | Naval Propulsion    | -10.01 ± 0.01 |    -8.00 ± 0.01 | −8.04 ± 0.04 |
> | Power Plant         |   2.77 ± 0.02 |     2.72 ± 0.02 |  2.66 ± 0.01 |
> | Wine Quality Red    |  -0.98 ± 0.06 |    -1.04 ± 0.05 | −0.77 ± 0.07 |
> | Yacht Hydrodynamics |   1.07 ± 0.27 |     1.08 ± 0.22 |  0.17 ± 0.25 |
>
> Comparison with heavy-tailed processes: the heavy-tailed stochastic processes (Favaro et al., 2020 and Bracale et al., 2021) do not admit closed-form posterior expression nor easy-to-implement posterior inference algorithm because the models presented there cannot easily be represented as a combination of simple models as ours.  Also, they need carefully chosen parametrization and nonlinearity functions to guarantee convergence. Hence, those methods are not applicable to our setting. We think that this fact highlights the strength of our method; compared with (Favaro et al., 2020 and Bracale et al., 2021), ours enjoys nice properties making efficient inference feasible, while supporting a broad class of practically-important stochastic processes. We also add that when the output is multi-dimensional, its dimensions become dependent in the limiting stochastic process of our work, while they become independent in the limit of  (Favaro et al., 2020 and Bracale et al., 2021).
>
> Typos: thanks. We have corrected them in the revised version.

---

> > ### Author Response · Authors · 2021-11-13
> > **Response to Reviewer Z24X  Continue**
> >
> >
> > (Favaro et al., 2020) Favaro, S., Fortini, S., and Stefano, P. Stable behaviour of infinitely wide deep neural networks. AISTATS, 2020.
> >
> > (Bracale et al., 2021) Bracale, D., Favaro, S., Fortini, S., and Peluchetti, S.  Infinite-channel deep stable convolutional neural networks. arXiv preprint arXiv:2102.03739.
> >
> > (Adlam et al., 2020) Adlam, B., Lee, J., Xiao, L., Pennington, J., & Snoek, J. Exploring the Uncertainty Properties of Neural Networks’ Implicit Priors in the Infinite-Width Limit. ICLR, 2021
> >
> > (Mukhoti et al., 2018) Mukhoti, J., Stenetorp, P., and Gal, Y. On the importance of strong baselines in bayesian deep learning. arXiv preprint arXiv:1811.09385.
> >
> > (Lakshminarayanan et al., 2017) Lakshminarayanan, B., Pritzel, A., and Blundell, C.  Simple and scalable predictive uncertainty estimation using deep ensembles. Advances in neural information processing systems, 2017.

---

> > ### Comment · Reviewer_Z24X · 2021-11-22
> > **Acknowledgement of Rebuttal**
> >
> > Thank you very much for replying to all reviews, and for running the additional experiments. I appreciated that you re-ran the experiment on the UCI datasets as suggested using identical splits for the competing method, as well as providing the missing RMSE results. The highlighted changes in the paper have also been noted.
> >
> > Although the overall verdict on the paper varies across the reviews, I remain of the opinion that there were several shortcomings in the original submission that would still benefit from an additional round of reviewing. I think that incorporating all of the suggested feedback in a new revision will only make the submission stronger, and thus encourage the authors to consider resubmitting a stronger version of this work. Nevertheless, in appreciation of the authors’ efforts in the rebuttal and the various clarifications that have been provided, I am raising my score to a 6.

---

### Official Review · Reviewer_axPe · 2021-11-02

**Correctness:** 4
**Technical Novelty And Significance:** 3
**Empirical Novelty And Significance:** 3
**Recommendation:** 8
**Confidence:** 5

**Main Review:**

Strengths: The approach is interesting, somewhat novel, and easily applicable to a broad range of deep nets. The provided theorems are sufficient for supporting the method and correct. The derivations pertaining to posterior inference are correct and result in efficient algorithms.  The results are broad enough, with many comparisons and many datasets considered; these vouch for the usefulness of the method.
Cons: Some discussion on complexity is missing.

**Summary Of The Paper:**

The paper extends upon the corpus of works that consider the infinite-width limit of a deep net. The foundation of the work is the Neural Network Gaussian Process (NNGP) model, which is taken as the infinite-width limit of a deep net under certain (mild) assumptions; that is the so-called Master Theorem.

The here proposed approach treats the variances, σ^2, of the penultimate layer weights as random variables. Then, it imposes an appropriate prior distribution on these and proceeds with a reiteration of the Master Theorem (and how this varies in this setting).

They show that, under this setting, the infinite-width limit of the network is a scale-mixture of NNGPs. In the special case of an imposed Gamma-prior, the so-obtained scale mixture reduces to a Student's-t process; this is a well-known result.



**Summary Of The Review:**

A strong paper lacking some discussion on complexity.

---

> ### Author Response · Authors · 2021-11-13
> **Response to Reviewer axPE**
>
> We thank you for your positive remarks on our work. For time complexity, as we mentioned in Section 3.1, our posterior-predictive algorithm based on importance sampling does not induce significant overhead thanks to the reuse of the shared terms for calculation. More specifically, our algorithm with K sample variances spends O(K+N^3) time, instead of O(KN^3), for computing a posterior predictive, where N is the number of training points. Compare this with the usual time complexity O(N^3) of the standard algorithm for Gaussian processes. When it comes to SVGP and SVTP, one update step of both SVGP and SVTP takes O(BM^2+M^3) time, where B is the number of the batch size of the input dataset and M is the number of the inducing points. We have added the analysis on time complexity in the appendix of the revised paper.

---

### Author Response · Authors · 2021-11-13
**Common response to the reviewers**

We thank all the reviewers for their constructive comments. We are happy to see the reviewers acknowledging the novelty of our theoretical extension. In the revision, we have added additional experimental results measuring the RMSE of the regression (appendix H, Table 5) and discussion on the time-complexity (appendix E.2). We also tried to proofread the paper again and fix some typos pointed out by the reviewers. We respond to the individual concerns raised by the reviewers in the responses below.

---

### Author Response · Authors · 2021-11-22
**The end of the discussion phase approaching**

Dear Reviewers,

Could you please go over our responses and the revision before the end of the author-reviewer discussion phase? We think we have faithfully responded to your comments and reflected them in the revision with some additional experiments, so we'd like to hear back from you whether they resolved your initial concerns. We sincerely thank you for your time and efforts in reviewing our paper, and your insightful and constructive comments.

Thanks, Authors

---

### Decision · Program_Chairs · 2022-01-20

**Decision:**

Accept (Poster)

**Comment:**

This paper presents a new formulation for the infinitely wide limiting case of deep networks as Gaussian processes, i.e. NNGPs.  The authors extend the existing case to incorporate a scale term at the penultimate layer of the network, which results in a scale mixture of NNGPs or a Student-t process in a specific case.  This formulation allows for a more heavy tailed output distribution which e.g. can be more robust to outliers.  The four reviews averaged just above borderline, with a 5, 8, 6, 6.  The reviewers found the approach to be sensible, technically correct and timely given the recent literature.  They found the experiments to be compelling for the most part, demonstrating the added robustness of this approach over the baseline NNGP.  The main concern raised by the reviewers is that the work is incremental, given that both NNGPs and Student-t processes are already established.